# Genetic, cellular, and structural characterization of the membrane potential-dependent cell-penetrating peptide translocation pore

Evgeniya Trofimenko[1], Gianvito Grasso[2†], Mathieu Heulot[1†], Nadja Chevalier[1†], Marco A Deriu[3], Gilles Dubuis[1], Yoan Arribat[1], Marc Serulla[1], Sebastien Michel[1], Gil Vantomme[4], Florine Ory[1], Linh Chi Dam[1], Julien Puyal[4,5], Francesca Amati[1], Anita Lüthi[4], Andrea Danani[2], Christian Widmann[1*]

[1]Department of Biomedical Sciences, University of Lausanne, Lausanne, Switzerland; [2]Dalle Molle Institute for Artificial Intelligence Research, Università della Svizzera italiana, Scuola Universitaria Professionale della Svizzera Italiana, Lugano, Switzerland; [3]PolitoBIOMed Lab Department of Mechanical and Aerospace Engineering, Politecnico di Torino, Torino, Italy; [4]Department of Fundamental Neurosciences, University of Lausanne, Lausanne, Switzerland; [5]CURML (University Center of Legal Medicine), Lausanne University Hospital, Lausanne, Switzerland

*For correspondence:
Christian.Widmann@unil.ch

†These authors contributed equally to this work

Competing interest: The authors declare that no competing interests exist.

**Abstract** Cell-penetrating peptides (CPPs) allow intracellular delivery of bioactive cargo molecules. The mechanisms allowing CPPs to enter cells are ill-defined. Using a CRISPR/Cas9-based screening, we discovered that KCNQ5, KCNN4, and KCNK5 potassium channels positively modulate cationic CPP direct translocation into cells by decreasing the transmembrane potential ($V_m$). These findings provide the first unbiased genetic validation of the role of $V_m$ in CPP translocation in cells. In silico modeling and live cell experiments indicate that CPPs, by bringing positive charges on the outer surface of the plasma membrane, decrease the $V_m$ to very low values (–150 mV or less), a situation we have coined megapolarization that then triggers formation of water pores used by CPPs to enter cells. Megapolarization lowers the free energy barrier associated with CPP membrane translocation. Using dyes of varying dimensions in CPP co-entry experiments, the diameter of the water pores in living cells was estimated to be 2 (–5) nm, in accordance with the structural characteristics of the pores predicted by in silico modeling. Pharmacological manipulation to lower transmembrane potential boosted CPP cellular internalization in zebrafish and mouse models. Besides identifying the first proteins that regulate CPP translocation, this work characterized key mechanistic steps used by CPPs to cross cellular membranes. This opens the ground for strategies aimed at improving the ability of cells to capture CPP-linked cargos in vitro and in vivo.

## Editor's evaluation

Although the role of membrane potential in cell-penetrating peptide (CPP) translocation has been consistently described in artificial systems, this multi scale study combining cell biology, genetics and in silico approaches further extends this topic to a live cell context where it shows that internalization stops when the membrane polarization is decreased by the removal of potassium channels. It proposes an original mechanism of CPP translocation based on transient water pore formation, which should be of interest for biophysicists, cell biologists and for applications such as drug deliver.

## Introduction

Cell-penetrating peptides (CPPs) are short non-toxic sequences of 5–30 amino acids present in proteins able to cross membranes such as homeoproteins and some viral components. CPPs can also be used to deliver bioactive cargos (siRNAs, DNA, polypeptides, liposomes, nanoparticles, and others) in cells for therapeutic or experimental purposes (*Bechara and Sagan, 2013*; *Futaki et al., 2013*; *Guidotti et al., 2017*; *Illien et al., 2016*; *Jones and Sayers, 2012*; *Koren and Torchilin, 2012*; *Madani et al., 2011*; *Mueller et al., 2008*; *Ruseska and Zimmer, 2020*; *Trabulo et al., 2010*; *Vasconcelos et al., 2013*). Even though they differ in their origin (*Frankel and Pabo, 1988*; *Green and Loewenstein, 1988*; *Joliot et al., 1991*; *Oehlke et al., 1998*) and physico-chemical properties, the majority of CPPs carry positive charges in their sequence (*Bechara and Sagan, 2013*; *Guidotti et al., 2017*; *Jones and Sayers, 2012*; *Madani et al., 2011*). Polyarginine (e.g. R9), HIV-1 TAT$_{47-57}$, and Penetratin (Antennapedia$_{43-58}$) are among the most used and studied CPPs.

The mode of CPP cellular entry is still debated and no proteins have been identified that regulate this process. CPP entry starts after the initial electrostatic interactions between the positively charged CPP and the negatively charged components of the cell membrane (*Bechara and Sagan, 2013*; *Futaki et al., 2013*; *Guidotti et al., 2017*; *Jones and Sayers, 2012*; *Koren and Torchilin, 2012*; *Madani et al., 2011*; *Ruseska and Zimmer, 2020*; *Trabulo et al., 2010*; *Vasconcelos et al., 2013*). Interaction with acid sphingomyelinase (*Verdurmen et al., 2010*) and glycosaminoglycans (*Amand et al., 2012*; *Bechara et al., 2013*; *Butterfield et al., 2010*; *Fuchs and Raines, 2004*; *Futaki and Nakase, 2017*; *Ghibaudi et al., 2005*; *Gonçalves et al., 2005*; *Hakansson and Caffrey, 2003*; *Rullo et al., 2011*; *Rusnati et al., 1999*; *Ziegler, 2008*; *Ziegler and Seelig, 2004*; *Ziegler and Seelig, 2011*), local membrane deformation (*Hirose et al., 2012*), as well as calcium fluxes (*Melikov et al., 2015*) have been suggested to play a role in CPP internalization. CPPs enter cells through a combination of two non-mutually exclusive mechanisms (*Illien et al., 2016*; *Bechara et al., 2013*): endocytosis and direct translocation (*Bechara and Sagan, 2013*; *Futaki et al., 2013*; *Guidotti et al., 2017*; *Jones and Sayers, 2012*; *Koren and Torchilin, 2012*; *Madani et al., 2011*; *Ruseska and Zimmer, 2020*; *Trabulo et al., 2010*; *Vasconcelos et al., 2013*). The nature of these entry mechanisms is debated and not fully understood at the molecular level. The vesicular internalization of CPPs has been suggested to occur through clathrin-dependent endocytosis, macropinocytosis, and caveolin-1-mediated endocytosis (*Bechara and Sagan, 2013*; *Futaki et al., 2013*; *Guidotti et al., 2017*; *Jones and Sayers, 2012*; *Koren and Torchilin, 2012*; *Madani et al., 2011*; *Trabulo et al., 2010*). However, recent data indicate that CPP endocytosis proceeds via a newly discovered pathway that is Rab14-dependent but Rab5- and Rab7-independent (*Trofimenko et al., 2021*). When CPPs are endocytosed, access to the cytosol requires that the CPPs break out of endosomes through a poorly understood process called endosomal escape.

Direct translocation allows the CPPs to access the cytosol through their ability to cross the plasma membrane. There is currently no unifying model to explain mechanistically how direct translocation proceeds and no genes have yet been identified to modulate the manner by which CPPs cross cellular membranes. Direct translocation across the plasma membrane often seemed to originate from specific areas of the cells, suggesting discrete structures on the plasma membrane involved in CPP entry (*Allolio et al., 2018*; *Duchardt et al., 2007*; *Hirose et al., 2012*; *Wallbrecher et al., 2017*; *Ziegler et al., 2005*). There is a general consensus though that an adequate plasma membrane potential ($V_m$) is required for direct translocation to occur based on live cell experiments (*Rothbard et al., 2004*; *Wallbrecher et al., 2017*; *Zhang et al., 2009*), as well as in silico studies (*Gao et al., 2019*; *Lin and Alexander-Katz, 2013*; *Moghal et al., 2020*; *Via et al., 2018*). Electrophysiological and pharmacological $V_m$ modulations have revealed that depolarization blocks CPP internalization (*Rothbard et al., 2004*; *Zhang et al., 2009*) and hyperpolarization improves the internalization of cationic CPPs (*Chaloin et al., 1998*; *Henriques et al., 2005*; *Moghal et al., 2020*; *Rothbard et al., 2004*; *Wallbrecher et al., 2017*). By itself, a sufficiently low $V_m$ (i.e. hyperpolarization) appears to trigger CPP direct translocation in live cells (*Rothbard et al., 2004*; *Wallbrecher et al., 2017*; *Zhang et al., 2009*). In silico modeling has provided evidence that membrane hyperpolarization leads to the formation of transient water pores, allowing CPP translocation into cells (*Gao et al., 2019*; *Herce and Garcia, 2007*; *Herce et al., 2009*; *Lin and Alexander-Katz, 2013*; *Via et al., 2018*), but the free energy landscape governing CPP translocation has not been determined. Moreover, the nature and

**eLife digest** Before a drug can have its desired effect, it must reach its target tissue or organ, and enter its cells. This is not easy because cells are surrounded by the plasma membrane, a fat-based barrier that separates the cell from its external environment. The plasma membrane contains proteins that act as channels, shuttling specific molecules in and out of the cell, and it also holds charge, with its inside surface being more negatively charged than its outside surface.

Cell-penetrating peptides are short sequences of amino acids (the building blocks that form proteins) that carry positive charges. These positive charges allow them to cross the membrane easily, but it is not well understood how.

To find out how cell-penetrating peptides cross the membrane, Trofimenko et al. attached them to dyes of different sizes. This revealed that the cell-penetrating peptides enter the cell through temporary holes called water pores, which measure about two nanometres across. The water pores form when the membrane becomes 'megapolarized', this is, when the difference in charge between the inside and the outside of the membrane becomes greater than normal. This can happen when the negative charge on the inside surface or the positive charge on the outer surface of the membrane increase. Megapolarization depends on potassium channels, which transport positive potassium ions outside the cell, making the outside of the membrane positive. When cell-penetrating peptides arrive at the outer surface of the cell near potassium channels, they make it even more positive. This increases the charge difference between the inside and the outside of the cell, allowing water pores to form. Once the peptides pass through the pores, the charge difference between the inside and the outside of the cell membrane dissipates, and the pores collapse.

Drug developers are experimenting with attaching cell-penetrating peptides to drugs to help them get inside their target cells. Currently there are several experimental medications of this kind in clinical trials. Understanding how these peptides gain entry, and what size of molecule they could carry with them, provides solid ground for further drug development.

the structural characteristics of the pores used by CPPs to cross the plasma membrane have not been investigated in live cells.

Here, we provide the first genetic evidence that validates the importance of $V_m$ for CPP direct translocation and we characterize the diameter of the water pores used by CPPs to enter live cells. We also determined the role of the $V_m$ in modulating the free energy barrier associated with membrane translocation and the impact of the $V_m$ on CPP translocation kinetics.

## Results

### Modes of TAT-RasGAP$_{317-326}$ cellular entry

In the present work, we have used TAT-RasGAP$_{317-326}$ as a model compound to investigate the molecular basis of CPP cellular internalization. This peptide is made up of the TAT$_{48-57}$ CPP and a 10 amino acid sequence derived from the SH3 domain of p120 RasGAP (*Michod et al., 2004*). TAT-RasGAP$_{317-326}$ sensitizes cancer cells to chemo-, radio-, and photodynamic therapies (*Chevalier et al., 2015*; *Michod et al., 2009*; *Pittet et al., 2007*; *Tsoutsou et al., 2017*) and prevents cell migration and invasion (*Barras et al., 2014*). This peptide also exhibits antimicrobial activity (*Heulot et al., 2017*; *Georgieva et al., 2021*; *Heinonen et al., 2021*). Some cancer cell lines, such as Raji (Burkitt's lymphoma), SKW6.4 (transformed B-lymphocytes), and HeLa (cervix carcinoma), are directly killed by this peptide (*Heulot et al., 2016*). The manner by which TAT-RasGAP$_{317-326}$ kills cells has recently been uncovered (*Serulla et al., 2020*). The peptide first accesses the cell's cytosol by direct translocation through the plasma membrane. It then binds to specific phospholipids, such as phosphatidylserine and phosphatidylinositol-bisphosphate that are enriched in the inner leaflet of the plasma membrane. This binding allows the peptide to disrupt the cell's membrane causing its death by necrosis.

Most CPPs can enter cells by direct translocation and by endocytosis (*Bechara and Sagan, 2013*; *Futaki et al., 2013*; *Guidotti et al., 2017*; *Illien et al., 2016*; *Jones and Sayers, 2012*; *Koren and Torchilin, 2012*; *Madani et al., 2011*; *Mueller et al., 2008*; *Ruseska and Zimmer, 2020*; *Trabulo et al., 2010*; *Vasconcelos et al., 2013*). This is also the case for TAT-RasGAP$_{317-326}$ (*Figure 1A–B* and

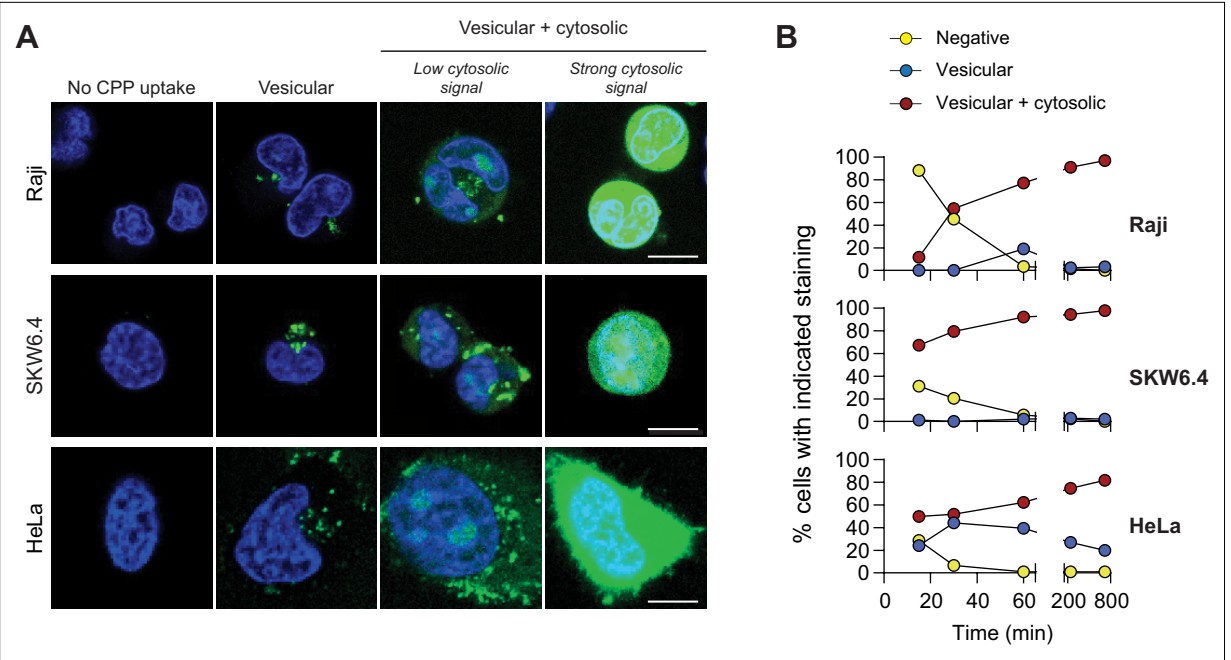

**Figure 1.** TAT-RaGAP$_{317-326}$ cellular entry modes. (**A**) Depiction of the different modes of cell-penetrating peptide (CPP) entry into cells. Confocal microscopy was performed on the indicated cell lines incubated for 1 hr with 40 µM FITC-TAT-RasGAP$_{317-326}$ in RPMI, 10 % fetal bovine serum (FBS). Cells were washed with PBS prior to visualization. Vesicular staining is indicative of CPP endocytosis while diffuse cytosolic staining is a consequence of CPP direct translocation into cells. Scale bar: 10 µm. (**B**) Quantitation of the different modes of CPP entry as a function of time (FITC-TAT-RasGAP$_{317-326}$ continually present in the media) using the experimental conditions presented in panel A. Types of staining were visually quantitated as indicated in *Figure 1—figure supplement 1A* (n = 157 cells per condition). There was no indication of fluorescence quenching, due to endosomal acidification, preventing the detection of CPP-containing endosomes (in at least during the first hour of CPP exposure) (*Figure 1—figure supplement 1B*). TAT-RasGAP$_{317-326}$ enters cells via endocytosis and direct translocation, but only direct translocation mediates its biological activity and leads to cell death (*Figure 1—figure supplement 4*). Results correspond to the average of three independent experiments.

The online version of this article includes the following figure supplement(s) for figure 1:

**Figure supplement 1.** TAT-RasGAP$_{317-326}$ internalization mode attribution and no impact of quenching for the detection of cell-penetrating peptide (CPP)-containing endosomes.

**Figure supplement 2.** No evidence of endosomal escape from TAT-RasGAP$_{317-326}$-containing endosomes.

**Figure supplement 3.** Cytosolic acquisition of fluorophore-labeled cell-penetrating peptides (CPPs) is not a consequence of laser-induced cellular damage.

**Figure supplement 4.** TAT-RasGAP$_{317-326}$ enters cells via endocytosis and direct translocation, but only direct translocation mediates its biological activity.

*Videos 1–3*). Two types of staining were observed in cells incubated with this peptide: (i) vesicular only and (ii) vesicular and cytosolic (*Figure 1A* and *Figure 1—figure supplement 1A*). When the peptide cytosolic signal was strong, it masked the vesicular staining (*Figure 1A*). In our experimental settings, the cytosolic acquisition of TAT-RasGAP$_{317-326}$ occurred only through direct translocation and not through endosomal escape (*Figure 1—figure supplement 2*, *Video 4*) and was not due to photo-toxicity (*Figure 1—figure supplement 3*) as can occur in some settings (*Dixit and Cyr, 2003*; *Ha and Tinnefeld, 2012*; *Levitus and Ranjit, 2011*; *Zheng et al., 2014*).

## Identification of potassium channels as mediators of TAT cargo direct translocation into cells

As TAT-RasGAP$_{317-326}$ needs to translocate through the plasma membrane to reach the cytosol, a prerequisite for the peptide to kill cells (*Serulla et al., 2020*), we used the killing ability of the peptide in a CRISPR/Cas9 screen to identify genes involved in CPP direct translocation in two different cell lines (Raji and SKW6.4 cells) (*Figure 2—figure supplement 1A*). The top candidate genes identified through this approach were specific potassium channels or genes coding for proteins known to

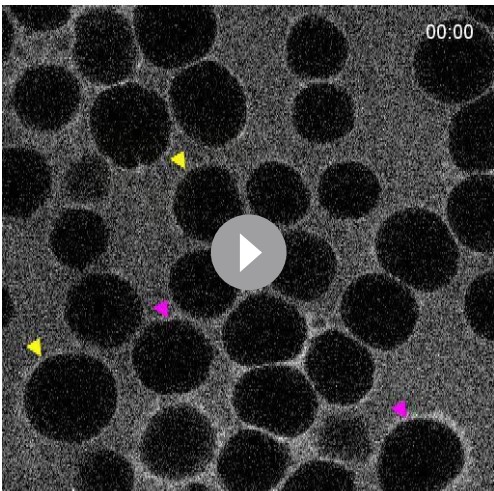

**Video 1.** TAT-RasGAP$_{317-326}$ internalization in Raji cells over a 16 -hr period. Representative confocal time-lapse recording of wild-type Raji cells incubated with 5 μM TAT-RasGAP$_{317-326}$ for 16  hr in RPMI in the absence of serum. For the first 30 min of the recording, images were taken every 30 s, then until the end of the recording, images were taken every 5 min. Peptide was present in the media throughout the recording. Yellow and pink arrows indicate cells taking up the peptide by direct translocation and by endocytosis, respectively. Cyan arrows point toward labeled endosomes and green asterisks to dead cells. Scale bar: 20 μm. Time is displayed in hours:minutes.

https://elifesciences.org/articles/69832/figures#video1

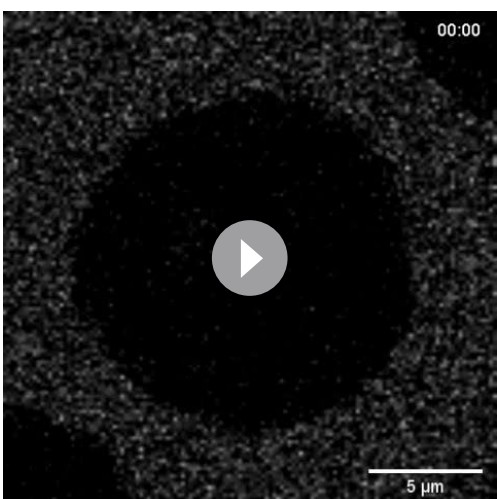

**Video 2.** Early peptide entry in wild-type Raji cells. Time-lapse recording of Raji cells incubated with 40 μM TAT-RasGAP$_{317-326}$ for 30 min in RPMI, 10 % fetal bovine serum (FBS). Peptide was present in the media throughout the recording and images were taken for 30 min at 10 s intervals. Scale bar: 10 μm. Time is displayed in minutes:seconds.

https://elifesciences.org/articles/69832/figures#video2

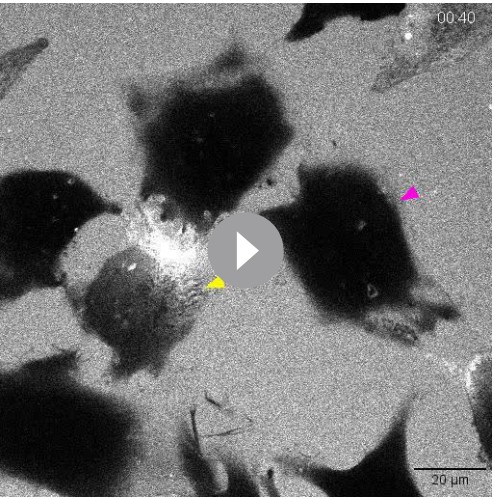

**Video 3.** Early peptide entry in wild-type HeLa cells. Time-lapse recording of HeLa cells incubated with 80 μM FITC-TAT-RasGAP$_{317-326}$ in RPMI, 10 % fetal bovine serum (FBS). Yellow and pink arrows indicate cells experiencing direct translocation and endocytosis, respectively. Images were taken for 30 min at 10 s intervals. Scale bar: 20 μm. Time is displayed in minutes:seconds.

https://elifesciences.org/articles/69832/figures#video3

regulate such channels indirectly (e.g. PIP5K1A; *Suh and Hille, 2008*; *Figure 2A* and *Figure 2— figure supplement 1B*). KCNQ5, identified in Raji cells, is a voltage-dependent potassium channel. KCNN4 and KCNK5, identified in SKW6.4 cells, are calcium-activated channels and belong to the two-pore (voltage-independent) potassium channel family (*Shieh et al., 2000*), respectively.

These potassium channels were pharmacologically or genetically inactivated (*Figure 2B* and *Figure 2—figure supplement 2A-C*) to validate their involvement in the direct translocation of

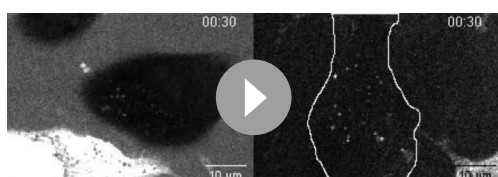

**Video 4.** Distinction between endosomal escape and direct translocation. Wild-type HeLa cells were pre-incubated with 80 μM FITC-TAT-RasGAP$_{317-326}$ for 30 min in RPMI, 10 % fetal bovine serum (FBS) and then imaged every 5 min for 4  hr at 37 °C, 5 % $CO_2$. Video on the left was recorded in the continuous presence of the peptide. Video on the right was recorded after the peptide was washed out three times with RPMI, 10 % FBS. Scale bar: 10 μm. Time is displayed in hours:minutes.

https://elifesciences.org/articles/69832/figures#video4

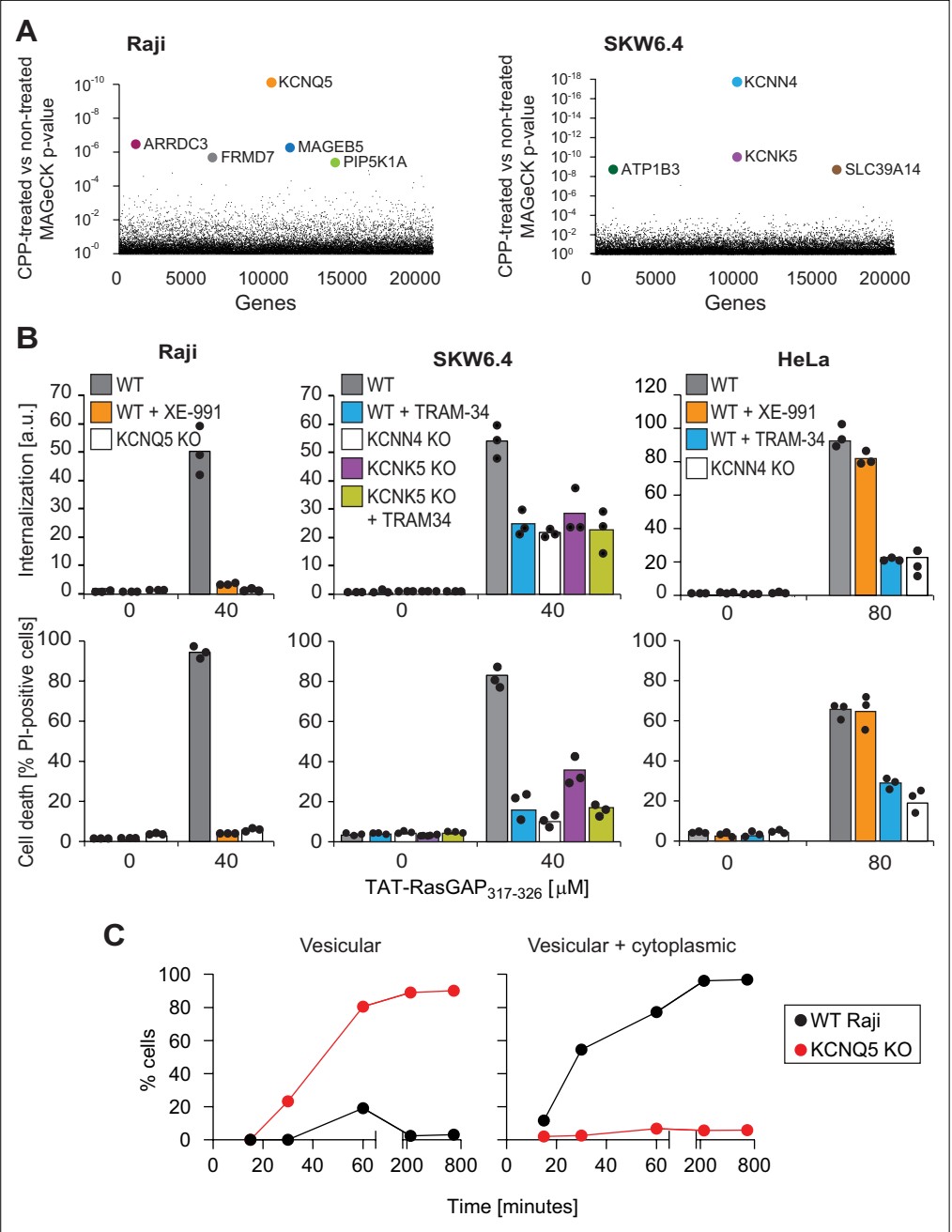

**Figure 2.** Identification of potassium channels as mediators of direct translocation of cell-penetrating peptides (CPPs) into cells. (**A**) Identification of genes implicated in TAT-RasGAP$_{317-326}$ internalization in Raji and SKW6.4 cells. The graphs depict the p-value (calculated using the MAGeCK procedure; see Materials and methods) for the difference in sgRNA expression between peptide-treated and control cells for the ~20,000 genes targeted by the CRISPR/Cas9 library. (**B**) Quantitation of TAT-RasGAP$_{317-326}$ entry (top) and induced death (bottom) in wild-type (WT) and knock-out (KO) cells. The WT and the corresponding potassium channel KO versions of the indicated cell lines were pretreated or not for 30 min with 10 μM XE-991 or with TRAM-34 and then incubated (still in the presence of the inhibitors when initially added) with or without 40 μM (Raji and SKW6.4 cells) or 80 μM (HeLa cells) TAT-RasGAP$_{317-326}$. Internalization was recorded after 1 hr and cell death after 16 hr (Raji and SKW6.4) or 24 hr (HeLa). Results correspond to the average of three independent experiments. TAT-RasGAP$_{317-326}$ concentrations and time of incubation used were adjusted so that the CPP induced similar cell death (between 60% and 90%) in the WT versions of the different cell lines. (**C**) Quantitation of the modalities of TAT-RasGAP$_{317-326}$ entry in WT and KCNQ5 KO Raji cells. Cells were incubated with FITC-TAT-RasGAP$_{317-326}$ for various periods of time and peptide staining was visually quantitated on confocal images (n = 165 cells for each time-point). The high percentage of cells with

*Figure 2 continued on next page*

*Figure 2 continued*

vesicular staining in the KO cells results from the absence of strong diffuse staining masking endosomes. The results correspond to the average of three experiments.

The online version of this article includes the following source data and figure supplement(s) for figure 2:

**Figure supplement 1.** CRISPR/Cas9 screen and target gene identification.

**Figure supplement 2.** Validation of the targets identified through the CRISPR/Cas9 screening.

**Figure supplement 2—source data 1.** Uncropped blot and original full raw unedited blot corresponding to the anti-FLAG blot shown in *Figure 2—figure supplement 2E*.

**Figure supplement 2—source data 2.** Uncropped blot and original full raw unedited blot corresponding to the anti-actin blot shown in *Figure 2—figure supplement 2E*.

**Figure supplement 2—source data 3.** Uncropped blot and original full raw unedited blot corresponding to the anti-V5 blot shown in *Figure 2—figure supplement 2F* (left-hand side).

**Figure supplement 2—source data 4.** Uncropped blot and original full raw unedited blot corresponding to the anti-$\alpha$-tubulin blot shown in *Figure 2—figure supplement 2F* (left-hand side).

**Figure supplement 2—source data 5.** Uncropped blot and original full raw unedited blot corresponding to the anti-V5 blot shown in *Figure 2—figure supplement 2F* (right-hand side).

**Figure supplement 2—source data 6.** Uncropped blot and original full raw unedited blot corresponding to the anti-tubulin blot shown in *Figure 2—figure supplement 2F* (right-hand side).

**Figure supplement 3.** Potassium channels modulate direct cell-penetrating peptide (CPP) translocation, but not endocytosis.

**Figure supplement 4.** Potassium channels regulate the cellular internalization of various TAT-bound cargos.

TAT-RasGAP$_{317-326}$ through the plasma membrane and the resulting death induction. The KCNQ family inhibitor, XE-991 (*Schroeder et al., 2000*), as well as KCNQ5 genetic invalidation (*Figure 2—figure supplement 2A*), fully blocked peptide internalization in Raji cells and protected them from the killing activity of the peptide (*Figure 2B* and *Figure 2—figure supplement 2D*). SKW6.4 cells individually lacking KCNN4 or KCNK5 (*Figure 2—figure supplement 2B*), or SKW6.4 cells treated with TRAM-34, a KCNN4 inhibitor (*Wulff et al., 2001*; *Wulff et al., 2000*), were impaired in their ability to take up the peptide and were partially protected against its cytotoxic activity (*Figure 2B* and *Figure 2—figure supplement 2D*). Inhibition of KCNN4 activity with TRAM-34 in KCNK5 knock-out cells did not further protect the cells against TAT-RasGAP$_{317-326}$-induced death. In HeLa cells, TRAM-34, but not XE-991, inhibited TAT-RasGAP$_{317-326}$ internalization and subsequent death (*Figure 2B*). Thus, in HeLa cells, KCNN4 channels regulate the membrane translocation of the peptide. This was confirmed by knocking out KCNN4 in these cells (*Figure 2B* and *Figure 2—figure supplement 2C*). Resistance to TAT-RasGAP$_{317-326}$-induced death in KCNQ5 knock-out Raji cells and KCNN4 knock-out SKW6.4 or HeLa cells was restored through ectopic expression of the corresponding FLAG- or V5-tagged channels (*Figure 2—figure supplement 2E-F*), ruling out off-target effects.

We next determined whether vesicular internalization or direct translocation were affected in cells with impaired potassium channel activities. Compared to their respective wild-type controls, the percentage of cells with diffuse cytosolic location of FITC-TAT-RasGAP$_{317-326}$ was drastically diminished in cells lacking one of the CRISPR/Cas9 screen-identified potassium channels in the respective cell lines (*Figure 2C* and *Figure 2—figure supplement 3A-B*). This was mirrored by an increase in the percentage of knock-out cells with vesicular staining. The invalidation of potassium channels did not affect transferrin or dextran internalization into cells (*Figure 2—figure supplement 3C*) or the infectivity of vesicular stomatitis virus (*Torriani et al., 2019*), substantiating the non-involvement of these channels in endocytic pathways.

One possibility to explain the above-mentioned results is that the absence of potassium channels reduces peptide binding to cells, thereby hampering subsequent peptide cellular uptake. At a 20 µM concentration, TAT-RasGAP$_{317-326}$ is readily taken up by wild-type Raji cells but not by KCNQ5 knock-out cells. At this concentration, peptide binding was slightly lower in knock-out than in wild-type cells (*Figure 2—figure supplement 3D*, upper graph). However, augmenting the peptide concentrations in the extracellular medium of KCNQ5 knock-out cells to reach surface binding signals equivalent or higher than what was obtained in wild-type cells still did not result in peptide cellular internalization

unless ≥80 µM of the peptides were used and even in this case, the uptake remained inefficient (*Figure 2—figure supplement 3D*). Difference in peptide binding is therefore not the cause of the inability of potassium channel knock-out cells to take up TAT-RasGAP$_{317-326}$.

We then assessed whether the role of potassium channels in cellular internalization also applied to TAT cargos other than RasGAP$_{317-326}$. TAT-PNA is an oligonucleotide covalently bound to TAT, which can correct a splicing mutation within the luciferase-coding sequence (*Abes et al., 2007*; *Kang et al., 1998*). This can only occur if TAT-PNA reaches the cytosol. The luciferase activity triggered by TAT-PNA was diminished in the presence of potassium channel inhibitors and in potassium channel knock-out cell lines (*Figure 2—figure supplement 4A*). Cytosolic access of TAT-Cre, which can recombine a loxP-RFP-STOP-loxP-GFP (*D'Astolfo et al., 2015*; *Wadia et al., 2004*) gene construct, was then assessed. Switch from red to green fluorescence occurs only when TAT-Cre reaches the nucleus. This took place in wild-type Raji cells but not in the KCNQ5 knock-out cells (*Figure 2—figure supplement 4B*). We finally tested a clinical phase III therapeutic D-JNKI1 compound (*Guidotti et al., 2017*; *Vasconcelos et al., 2013*) used in the context of hearing loss and intraocular inflammation. The internalization of this peptide was completely blocked in Raji cells lacking KCNQ5 (*Figure 2—figure supplement 4C*, left). D-JNKI1 internalization was also diminished in SKW6.4 cells lacking KCNN4 and KCNK5 channels, as well as in HeLa cells lacking the KCNN4 potassium channel (*Figure 2—figure supplement 4C*, middle and right panels). These data demonstrate that the absence of specific potassium channels diminishes or even blocks the entry of various TAT-bound cargos.

## Potassium channels maintain plasma membrane polarization that is required for cationic CPP entry into cells

Potassium is the main ion involved in setting the plasma membrane potential ($V_m$). The potassium channels identified in the CRISPR/Cas9 screen may therefore participate in the establishment of an adequate $V_m$ permissive for CPP direct translocation (*Chaloin et al., 1998*; *Henriques et al., 2005*; *Moghal et al., 2020*; *Rothbard et al., 2004*; *Wallbrecher et al., 2017*; *Zhang et al., 2009*). *Figure 3A* (left graph) shows that genetic disruption or pharmacological inhibition of KCNQ5 in Raji cells led to an increase in their $V_m$ (from –26 to –15 mV, validated with electrophysiological recordings; see *Figure 3—figure supplement 1A*). Surprisingly, such minimal increase in $V_m$ in the KCNQ5 knock-out Raji cells practically abolished CPP internalization (*Figure 3B*, left graph), indicating that above a certain threshold, the $V_m$ is no longer permissive for CPP direct translocation. In SKW6.4 and HeLa cells, $V_m$ measurement was much more variable than in Raji cells. Nevertheless, a trend of increased $V_m$ was observed when KCNN4 or KCNK5 were invalidated genetically or pharmacologically (*Figure 3A*, middle and right graphs) and this was accompanied by reduced peptide uptake (*Figure 3B*, middle and right graphs). As the CRISPR/Cas9 screens performed in various cell lines identified a variety of potassium channels required for efficient CPP internalization, we conclude that it is the $V_m$ maintenance activity of these channels that is important for CPP direct translocation and not some specific features of the channels.

If the reason why invalidation of the KCNQ5, KCNN4, and KCNK5 potassium channels inhibits TAT-RasGAP$_{317-326}$ cellular entry is cell depolarization, a similar response should be obtained by artificially depolarizing cells. Indeed, depolarizing cells with gramicidin (*Eisenman et al., 1978*) (making non-specific 0.4 nm pores [*Kelkar and Chattopadhyay, 2007*] in cell membranes) or by increasing the extracellular concentration of potassium (dissipating the potassium gradient) totally blocked cytosolic peptide acquisition into the three studied cell lines (*Figure 3B*) but not peptide endocytosis (*Figure 3—figure supplement 1B*). Hence, cellular depolarization in itself inhibits TAT-RasGAP$_{317-326}$ direct translocation into the cytosol.

Next, we determined whether hyperpolarization could reverse the inability of potassium channel knock-out cells to take up TAT-RasGAP$_{317-326}$. Cells were either incubated in the presence of valinomycin (*Rimmele and Chatton, 2014*), which leads to formation of potassium-like channels, or transfected with KCNJ2 channel that also provokes potassium efflux and membrane hyperpolarization (*Xue et al., 2014*). *Figure 3B* shows that hyperpolarization of cells lacking CRISPR/Cas9-identified potassium channels fully restored peptide translocation. Moreover, hyperpolarization increased peptide cytosolic acquisition in wild-type cells (*Figure 3B*). Similar effect, albeit to a lesser extent, was observed by ectopically expressing KCNQ5 in wild-type and KCNN4 knock-out SKW6.4 and HeLa cells as well as by ectopically expressing KCNN4 in wild-type and KCNQ5 knock-out Raji

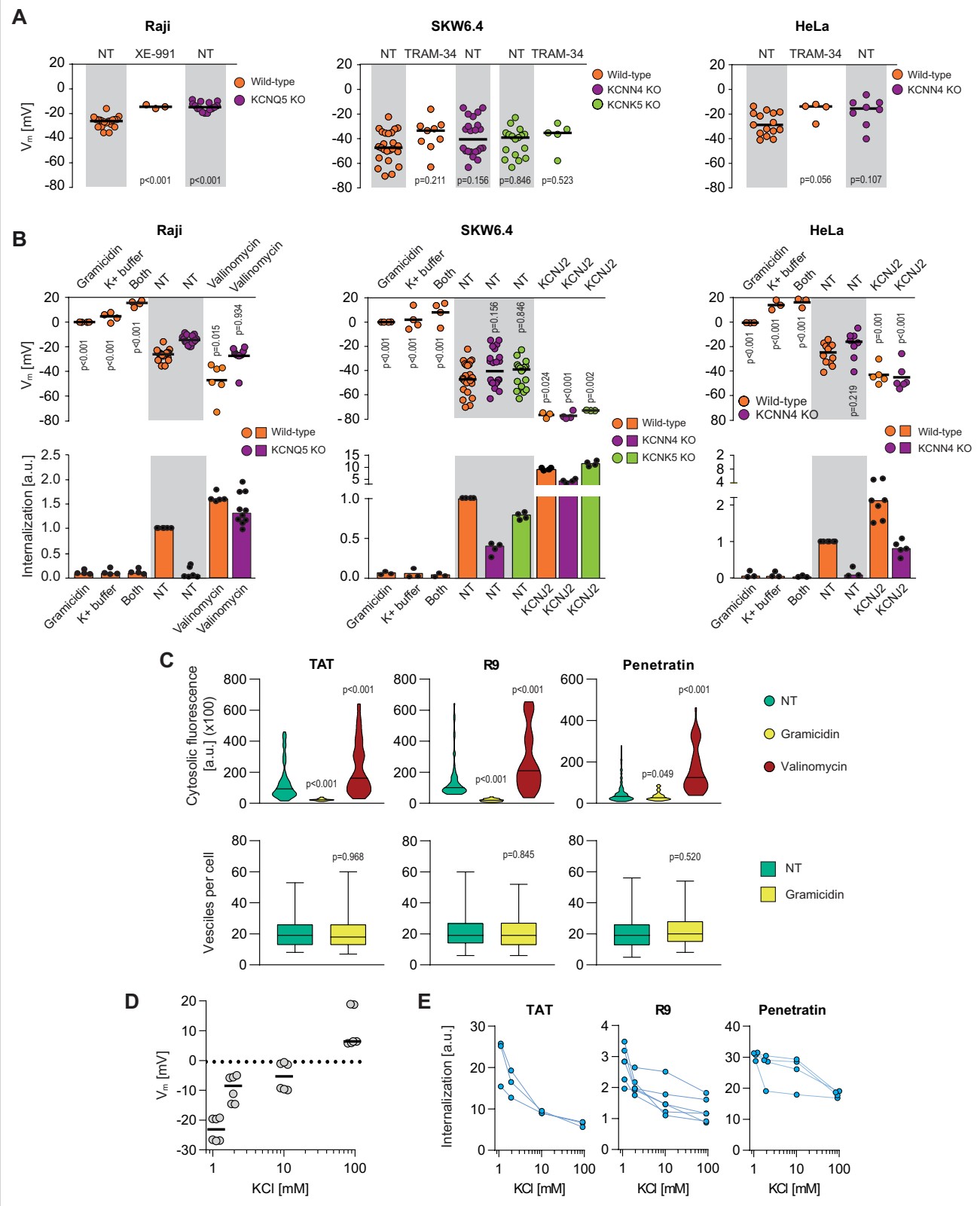

**Figure 3.** Potassium channels maintain plasma membrane polarization that is required for cell-penetrating peptide (CPP) entry into cells. (**A**) Assessment of the resting plasma membrane potential in the indicated wild-type cell lines and the corresponding potassium channel knock-out (KO) clones in the presence or in the absence 10 µM XE-991 or TRAM-34. The gray and white zones correspond to non-treated cells and inhibitor-treated cells, respectively. NT, not treated. The p-values correspond to the assessment of the significance of the differences with the control wild-type condition

*Figure 3 continued on next page*

*Figure 3 continued*

using ANOVA multiple comparison analysis with Dunnett's correction. Each dot in a given condition represents an independent experiment. (**B**) Effect of cellular depolarization (left of the gray zone) and hyperpolarization (right of the gray zone) on peptide internalization in the absence of serum. The indicated cell lines and the corresponding channel KO clones were pretreated or not with depolarization agents (2 µg/ml gramicidin for 5 min or high extracellular potassium buffer for 30 min) or with hyperpolarization inducer (10 µM valinomycin), followed by the addition of TAT-RasGAP$_{317-326}$ for 1 hr. Alternatively, hyperpolarization was achieved by ectopic expression of the KCNJ2 potassium channel. Membrane potential and peptide internalization were then determined. Membrane potential was measured in the presence of DiBac4(3) by flow cytometry. Peptide internalization was measured by flow cytometry in the presence of 0.2 % trypan blue. The p-values correspond to the assessment of the significance of the differences with the control wild-type condition using ANOVA multiple comparison analysis with Dunnett's correction. Each dot in a given condition represents an independent experiment. Treatment with valinomycin was used in the absence of serum as the latter is expected to interfere with the drug (*Rimmele and Chatton, 2014*). As shown in *Figure 3—figure supplement 5A*, removing serum from the culture medium sensitized cells to TAT-RasGAP$_{317-326}$ and consequently, the CPP concentration had to be adapted accordingly (*Figure 3—figure supplement 5B*). Serum withdrawal does not affect the V$_m$ (*Figure 3—figure supplement 5C*). (**C**) Quantitation of cytosolic CPP signal (top) and the number of endocytic vesicles per cell (bottom) in wild-type HeLa cells (n = 158 cells) incubated for 1 hr with 10 µM FITC-CPP in control, depolarizing (2 µg/ml gramicidin), or hyperpolarizing (10 µM valinomycin) conditions in the absence of serum based on confocal microscopy images (*Figure 3—figure supplement 2D*). Comparison between different conditions to non-treated control was done using ANOVA test with Dunnett's correction for multiple comparison. The number of endocytic vesicles per cell was quantitated based on confocal images. Statistical comparison was done using t-tests. Quantitation of vesicles was not performed in hyperpolarizing conditions due to masking from strong cytosolic signal. (**D**) Modulation of the V$_m$ membrane potential by varying extracellular potassium concentrations. Assessment of membrane potential changes in Raji cells incubated in RPMI medium containing the indicated concentrations of potassium chloride (isotonicity was maintained by adapting the sodium chloride concentrations; see Materials and methods). Membrane potential was measured with DiBac4(3). The results correspond to the median of six independent experiments. (**E**) Internalization of various CPPs in the presence of different concentrations of potassium chloride in the media. Data for a given experiment are linked with thin blue lines.

The online version of this article includes the following figure supplement(s) for figure 3:

**Figure supplement 1.** Importance of the V$_m$ for TAT-RasGAP$_{317-326}$ direct translocation.

**Figure supplement 2.** Importance of the V$_m$ for the direct translocation of several cationic cell-penetrating peptides (CPPs).

**Figure supplement 3.** Effect of V$_m$ modulation on TAT-RasGAP$_{317-326}$ translocation in primary neuronal cells.

**Figure supplement 4.** Cell-penetrating peptide (CPP) binding to cells is not affected by depolarization.

**Figure supplement 5.** Effect of serum on the sensitivity to TAT-RasGAP$_{317-326}$ and cell-penetrating peptide (CPP) internalization.

cells (*Figure 3—figure supplement 1C*). Additionally, cells such as primary rat cortical neurons that naturally have a low V$_m$ (–48 mV) take up the CPP in their cytosol more efficiently than cells with higher V$_m$ such as HeLa cells (–25 mV) (*Figure 3—figure supplement 1D*). Altogether, these results demonstrate that the V$_m$ modulates internalization of TAT-RasGAP$_{317-326}$ in various cell lines. This internalization can be manipulated through cellular depolarization to block it and through hyperpolarization to increase it, confirming earlier results obtained for the R8 CPP in Jurkat cells (*Rothbard et al., 2004*).

We then assessed whether the entry of TAT, nanomeric arginine (R9), and Penetratin (*Figure 3—figure supplement 2A*), three commonly used cationic CPPs in biology and medicine, was regulated by the plasma membrane potential as shown above for TAT-RasGAP$_{317-326}$. Similarly to TAT-RasGAP$_{317-326}$, these CPPs are taken up by HeLa cells by both direct translocation and endocytosis (*Figure 3—figure supplement 2B-C*). Depolarization, induced by either gramicidin or high extracellular potassium concentrations (*Figure 3D*), led to decreased cytosolic fluorescence of these CPPs, while valinomycin-mediated hyperpolarization favored their translocation in the cytosol (*Figure 3C*, upper graphs, *Figure 3E*, and *Figure 3—figure supplement 2D-E*). Although the cellular membrane composition of neurons may differ from the other cell lines used in this study, the V$_m$ also controlled peptide translocation in non-transformed rat primary cortical neurons (*Figure 3—figure supplement 3*). In contrast, depolarization had no impact on the ability of the cells to endocytose these CPPs (*Figure 3C*, bottom graphs), further confirming that CPP endocytosis is not affected by V$_m$. Finally, we note that CPP membrane binding was only minimally affected by depolarization (*Figure 3—figure supplement 4*). Hence, the reason why depolarized cells do no take up CPPs is not a consequence of reduced CPP binding to cells, confirming our earlier observation obtained with TAT-RasGAP$_{317-326}$ (*Figure 2—figure supplement 3D*). Altogether the data presented in *Figure 3* show that direct translocation of cationic CPPs is modulated by the V$_m$ of cells and that specific potassium channels are involved in this modulation.

## CPP direct translocation modeling

To further study the mechanism of CPP cellular entry through direct translocation, we took advantage of coarse-grained molecular dynamics (MD) technique and MARTINI force field 2.2 p (*Marrink et al., 2007*; *Marrink and Tieleman, 2013*). In our simulations we have used TAT-RasGAP$_{317-326}$, TAT, R9 and Penetratin in presence of a natural cell membrane-like composition (for both inner and outer leaflets) while earlier studies have employed simpler membrane composition (*Gao et al., 2019*; *Herce et al., 2009*; *Lin and Alexander-Katz, 2013*; *Moghal et al., 2020*; *Via et al., 2018*; *Zhang et al., 2009*). Membrane hyperpolarization was achieved by setting an ion imbalance (*Delemotte et al., 2008*; *Gao et al., 2019*; *Gurtovenko and Vattulainen, 2007*; *Herrera and Pantano, 2009*) through a net charge difference of 30 positive ions (corresponding to a $V_m$ of ~2 V) between the intracellular and extracellular spaces. The use of very high $V_m$ values, typically used in computational studies, is required to capture nanosecond occurring events. This protocol (*Figure 4—figure supplement 1A*) allowed us to observe CPP translocation across membranes within a few tens of nanoseconds (*Figure 4A* and *Video 5*). In presence of ~2 V $V_m$, the CPPs approached the membrane on the extracellular side and this led to the formation of a water column within the membrane that the CPP then used to move to the intracellular space (*Video 5*). The movement of the positive charges carried by the CPPs, as well as extracellular cations, to the intracellular compartment via the water pore induced membrane depolarization. This depolarization provoked the collapse of the water pore and membrane resealing. Even though CPPs play an active role in their internalization, the mere presence of the CPP in the absence of a sufficiently low $V_m$ was not enough to trigger water pore formation (*Figure 4B*, right graph and *Video 6*). These data confirm earlier work describing the role of the $V_m$ in CPP penetration into or through bilipidic membranes (*Gao et al., 2019*; *Herce and Garcia, 2007*; *Herce et al., 2009*; *Lin and Alexander-Katz, 2013*; *Moghal et al., 2020*; *Via et al., 2018*; *Zhang et al., 2009*). TAT, R9, and Penetratin all translocated into the intracellular compartment but with different propensities (*Figure 4B*, left graph) and with different kinetics (*Figure 4—figure supplement 1B*) that appeared to be related to the positive charges they carry (*Figure 3—figure supplement 2A*): the more positively charged a CPP, the higher probability to translocate across cell membranes and the faster kinetics of water pore formation at a given $V_m$.

We also applied a metadynamics protocol to estimate the impact of the $V_m$ on the free energy landscape of R9 translocation. The free energy barriers recorded in depolarized membranes ($V_m = 0$) and polarized membranes ($V_m = -80$ mV) were similar (*Figure 4C–D*). The obtained value of about 200 kJ/mol is in line with recent estimation of the free energy barrier associated with CPP translocation at a $V_m = 0$ (*Gao et al., 2019*). Only at much lower $V_m$ values (–150 mV) was a marked decrease in free energy barrier recorded. This indicates that hyperpolarization values found in resting cells (down to about –80 mV in neurons and higher in many other cells types; *Yang and Brackenbury, 2013*) are not more favorable than fully depolarized membranes to establish conditions for the formation of water pores. It appears therefore that cells need to decrease their $V_m$ to much lower values (e.g. –150 mV or lower) to reach conditions compatible with water pore formation. This in silico observation may appear contradictory with our results obtained in live cells showing direct translocation at –25 mV (*Figure 2*), as well as with the experiment demonstrating that CPP cytosolic internalization was more efficient in cortical neurons in comparison to less negatively charged HeLa cells (*Figure 3—figure supplement 1D*). We therefore postulate that the presence of CPPs on the cell surface induces locally a substantial voltage drop from the resting $V_m$. To test this assumption, we analyzed the electrostatic potential map in a molecular system composed of the R9 peptide in contact with the plasma membrane in the absence of an external electrostatic field (*Figure 4E*). This simulation indicated that the presence of CPPs at the cell surface is sufficient to decrease locally the transmembrane potential to about –150 mV (*Figure 4E*). This was not observed in the absence of the CPP. In conclusion, our data support a model where CPPs further decrease the $V_m$ of resting cells to very low values (equal to or less than –150 mV) that are compatible with spontaneous water pore formation and that we coin megapolarization.

Our model also predicts that the electric force exerted on CPPs when cells are megapolarized permit CPPs to accumulate in the cytosol and reach concentrations that are higher than in the extracellular milieu. *Figure 4—figure supplement 1C* shows indeed that cells can concentrate TAT-RasGAP$_{317-326}$ in the cytosol of Raji and HeLa cells, up to 100 times in extreme cases.

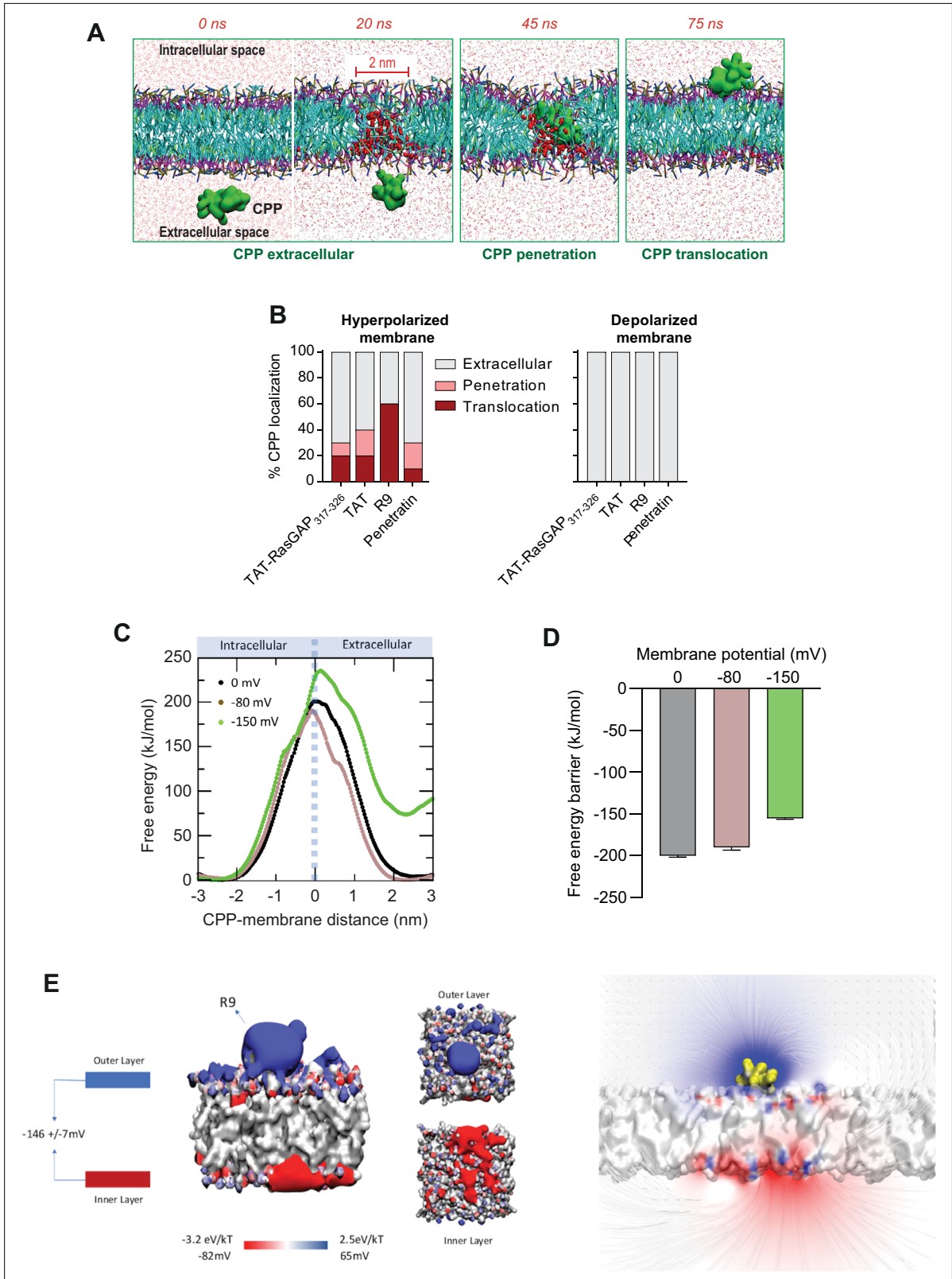

**Figure 4.** Hyperpolarization favors the formation of ~2-nm-wide water pores used by cell-penetrating peptides (CPPs) to translocate into cells. (**A**) Visualization of in silico modeled, time-dependent, TAT-RasGAP$_{317-326}$ penetration and subsequent translocation across cellular membrane through a water pore. Water molecules within membranes are depicted by red spheres (and by red dots outside the membrane). (**B**) Quantitation of CPP localization in hyperpolarized or depolarized conditions based on coarse-grained molecular dynamics simulations. Membrane hyperpolarization was

*Figure 4 continued*

achieved through a net charge difference of 30 positive ions between intracellular and extracellular space in a double-bilayer system (*D'Astolfo et al., 2015*, *Kang et al., 1998*; *Kauffman et al., 2018*; *Kelkar and Chattopadhyay, 2007*; *Khalil et al., 2018*) obtaining a transmembrane potential of –2.2 V. Such low membrane potential was required to visualize translocation within the time frame of the simulations (100 ns). (**C**) Free energy landscape of R9 translocation reported as a function of CPP-membrane distance. The metadynamics simulations were performed at transmembrane potential values of 0 , –80 , and –150 mV (black, brown, and green curves). (**D**) Free energy barrier for CPP translocation at different transmembrane potential values. (**E**) Electrostatic potential map of a molecular system containing one R9 peptide in contact with the cell membrane, without any applied external electrostatic field.

The online version of this article includes the following figure supplement(s) for figure 4:

**Figure supplement 1.** In silico modeling of cell-penetrating peptide (CPP) direct translocation.

## Structural characterization of the pore allowing CPP entry in live cells

Propidium iodide (PI), with a diameter of 0.8–1.5 nm (*Bowman et al., 2010*) or fluorophore-labeled 3 , 10 , and 40 kDa dextrans, with diameters (provided by Thermo Fisher) of 2.3 ± 0.38 (*Thorne and Nicholson, 2006*), 4.5,  and 8.6 nm, respectively (*Figure 5—figure supplement 1A*), were used to estimate the size of the water pores formed in the presence of CPPs in live cells. These molecules by themselves did not translocate in the cytosol of cells (*Figure 5A* and *Figure 5—figure supplement 1B*). They were then co-incubated with different FITC-labeled CPPs and their uptake monitored by confocal microscopy. While PI and CPPs efficiently co-entered cells (*Figure 5B* and *Figure 5—figure supplement 1C-D*), there was only marginal co-entry of the dextrans with the CPPs (*Figure 5B*). The marginal dextran co-entry was inversely correlated with the dextran diameters (inset in *Figure 5B*): ~2.3-nm-wide dextrans entered cells better than ~4.5-nm-wide dextrans and ~8.6-nm-wide dextrans mostly remained outside cells. The entry of PI and CPPs in cells occurred with identical kinetics (*Figure 5—figure supplement 1D*), further supporting the notion that they enter cells together. The PI/CPP co-entry was prevented by cell depolarization (*Figure 5—figure supplement 1B*), which is expected if PI accesses the cytosol via the megapolarization-induced pores used by CPP to enter cells. CPPs do not need to be labeled with a fluorophore to allow PI co-entry into cells (*Figure 5B*, 'unlabeled TAT+ PI' condition), ruling out phototoxicity as a confounding effect. Similar results were obtained in primary rat cortical neurons, where PI cytosolic signal was observed in cells that took up the selected CPPs through direct translocation (*Figure 5C*). These data are compatible with the notion that water pores triggered by CPPs allow molecules up to ~2 nm in diameter to efficiently enter cells. They are also in line with the in silico prediction of the water pore diameter of 1.6±0.26 nm obtained by analyzing the structure of the pore at the transition state (i.e. when the CPP is crossing the cell membrane; see *Figure 4A*). Molecules in the 2–5 nm diameter range, such as 3 and 10 kDa dextrans, can still use this entry route to a limited extent.

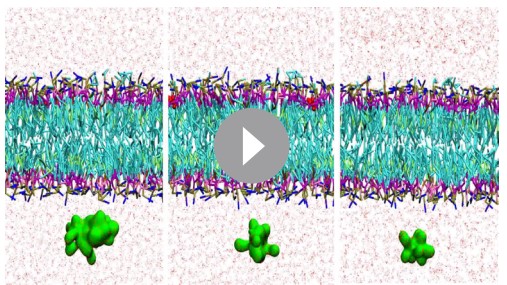

**Video 5.** In silico visualization of water pore formation in the presence of the indicated cell-penetrating peptides (CPPs) across a polarized membrane bilayer. This video shows the translocation of the indicated CPPs across a plasma membrane in the presence of a membrane potential of –2.2 V. This simulation was performed by molecular dynamics MARTINI coarse-grained approach using an asymmetric multi-component bilayer in the presence of ion imbalance to polarize the membrane.

https://elifesciences.org/articles/69832/figures#video5

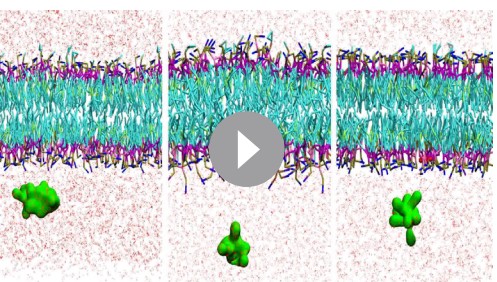

**Video 6.** In silico visualization of water pore formation in the presence of the indicated cell-penetrating peptides (CPPs) across a non-polarized membrane bilayer. This video shows the lack of translocation of the indicated CPPs across a plasma membrane in the absence of a membrane potential (0 V). This simulation was performed by molecular dynamics MARTINI coarse-grained approach using an asymmetric multi-component bilayer in the absence of ion imbalance.

https://elifesciences.org/articles/69832/figures#video6

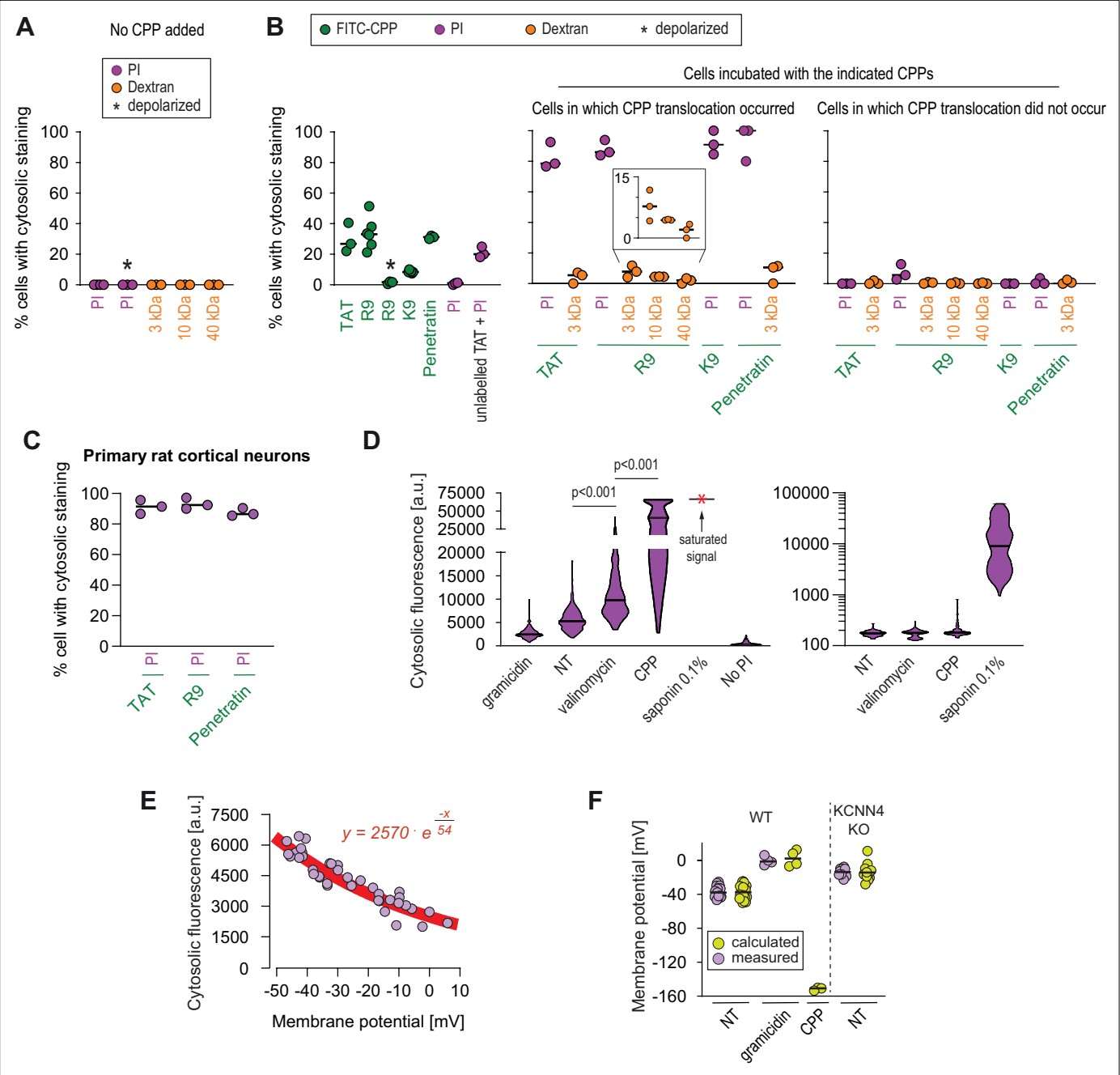

**Figure 5.** Estimation of the size of the pore used by cell-penetrating peptides (CPPs) to enter cells. (**A–B**) Quantitation of the percentage of cells with cytosolic staining after the indicated treatment. The indicated compounds (32 µg/ml propidium iodide [PI], 200 µg/ml dextran, 40 µM CPP) were incubated for 30 min with HeLa cells. Depolarization, indicated by an asterisk, was induced with 2 µg/ml gramicidin. The percentage of cells displaying cytosolic internalization of the indicated molecules was then determined on confocal images (n = 207 cells; see the Materials and methods and *Figure 5—figure supplement 1C*). Inset corresponds to an enlargement of the percentage of cells positive for dextran in the presence of R9. The results correspond to at least three independent experiments. CPPs such as R9 do not bind to PI (*Figure 5—figure supplement 2A*) and thus PI entry and accumulation within cells was not the result of CPP carry over. (**C**) Quantitation of the percentage of primary rat cortical neurons with cytosolic staining following incubation for 30 min with the indicated CPPs (2 µM) and PI (32 µg/ml). The percentage of cells displaying cytosolic internalization of the indicated molecules was then determined on confocal images (n = 153 cells), as in panel B. (**D**) Left graph: quantitation of PI cytosolic internalization in wild-type HeLa cells after 30 min of incubation in normal, depolarizing (2 µg/ml gramicidin) or hyperpolarizing (10 µM valinomycin) conditions in the presence or in the absence of 40 µM FITC-R9. Right graph: as in left graph, but using lower laser power to avoid saturation of the signal obtained in saponin-permeabilized cells. Cytosolic internalization was quantitated from confocal images using ImageJ (n = 319 cells; see Materials and methods). The p-values correspond to the assessment of the significance of the differences with the non- treated (NT) control condition using ANOVA multiple

*Figure 5 continued on next page*

*Figure 5 continued*

comparison analysis with Dunnett's correction. The results correspond to three independent experiments. PI staining is commonly used to assess cell membrane integrity, frequently associated with cell death (see for example *Figure 2B*, lower graphs). This dye poorly fluoresces in solution (*Figure 5—figure supplement 2B*). However, the PI cytosolic intensity values in dead permeabilized cells are several orders of magnitude higher than those recorded after cell hyperpolarization (compare the left and right graphs in the present panel). (**E**) Relation between cytosolic PI intensity and membrane potential measured with the DiBac4(3) sensor in HeLa cells. Each dot represents an independent experiment. (**F**) The fitted curve from panel E was used to calculate membrane potential values based on cytosolic PI intensities in HeLa cells and its corresponding KCNN4 knock-out (KO). These values are those labelled "calculated" in the graph. Those labelled "measured" correspond to the membrane potentials determined via DiBac4(3) uptake. Each dot in a given condition represents an independent experiment.

The online version of this article includes the following figure supplement(s) for figure 5:

**Figure supplement 1.** Evidence for low molecular weight pore formation in living cells during cell-penetrating peptide (CPP) direct translocation.

**Figure supplement 2.** Control experiments pertinent to the use of propidium iodide (PI).

In this context, the Cre recombinase, with a diameter of 5 nm (estimated from its crystal structure; NDB:PD0003), can be transported by TAT into cells (*Figure 2—figure supplement 4B*), another indication that the pores used by cationic CPPs to enter cells can allow the passage of molecules up to 5 nm.

Despite identical net positive charges (*Figure 3—figure supplement 2A*), and as reported earlier (*Mitchell et al., 2000*), the K9 peptide made of nine lysine residues was less capable of translocating into cells compared to R9 (*Figure 5B* and *Figure 3—figure supplement 2C*, right graph). This may be due to the deprotonation of K9 once in the plasma membrane (see Discussion). However, in the few cases when cells have taken up K9, PI co-internalized as well (middle graph of *Figure 5B*). This indicates that K9 has a reduced capacity compared to R9 to trigger water pore formation but when they do, PI can efficiently translocate through the pores created by K9.

Modeling experiments indicate that water pores are created in membranes subjected to sufficiently high (absolute values) $V_m$. We therefore tested whether the mere hyperpolarization of cells (i.e. in the absence of CPPs) could trigger the translocation of PI into cells, indicative of water pore formation. *Figure 5D* (left) shows that the hyperpolarizing drug valinomycin significantly increased PI cell permeability. In contrast, depolarization, mediated by gramicidin, reduced PI internalization (*Figure 5D*, left). Cells incubated with CPPs took up PI in their cytosol to a much greater extent than when cells were treated with valinomycin (*Figure 5D*, left), as expected if CPPs participate in setting plasma membrane megapolarization.

*Figure 5E* shows the correlation between cytosolic PI accumulation over time and $V_m$. Based on this correlation, we estimated the $V_m$ of cells incubated with a CPP to be in the order of –150 mV (*Figure 5F*). In accordance with the modeling experiments, these data further support the notion (i) that water pore formation in cells is favored by cell hyperpolarization and inhibited by depolarization and (ii) that CPPs themselves (*Rao et al., 2014*; *Wallbrecher et al., 2017*) further contribute to the establishment of local megapolarization in the plasma membrane.

## Megapolarization improves CPP internalization in vivo

We investigated whether it was possible to experimentally manipulate the $V_m$ to favor CPP internalization in in vivo situations. Systemic exposure of zebrafish embryos to valinomycin in Egg water led to cell hyperpolarization (*Figure 6—figure supplement 1A*) and improved internalization of a TAT-based CPP (*Figure 6—figure supplement 1B*). This systemic treatment, while not acutely toxic, halted development (*Figure 6—figure supplement 1C-E*). However, local valinomycin injection did not affect long-term viability (*Figure 6—figure supplement 1F*) and efficiently increased CPP cellular internalization (*Figure 6A*). Subcutaneous injections of valinomycin in mice induced tissue hyperpolarization (*Figure 6—figure supplement 1G*) and boosted the CPP delivery in skin cells (*Figure 6B*). These results demonstrate that hyperpolarizing drugs can be used to ameliorate CPP internalization in animal tissues.

## Discussion

Multiple models, mostly inferred from artificial experimental paradigms, have been proposed to explain CPP direct translocation. These include the formation of pores made of the CPPs themselves

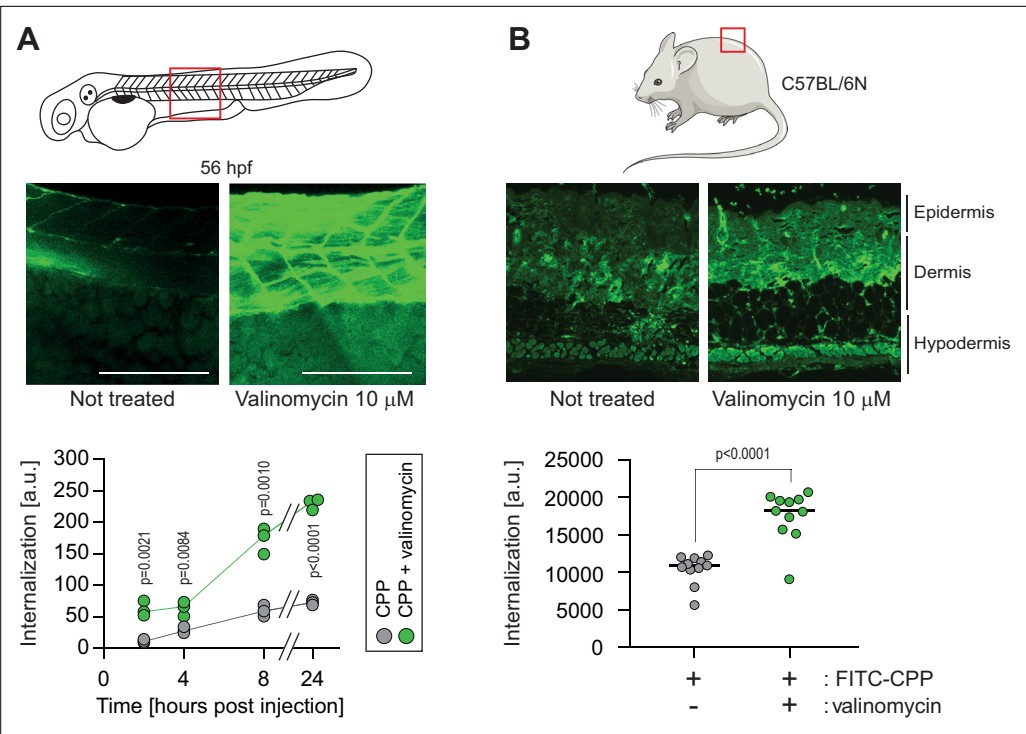

**Figure 6.** Hyperpolarization improves cell-penetrating peptide (CPP) internalization in vivo. (**A**) CPP internalization in zebrafish embryos in normal and hyperpolarized conditions. Forty-eight-hour post fertilization, zebrafish embryos were injected with 3.12 µM FITC-TAT-RasGAP$_{317-326}$(W317A) with or without 10 µM valinomycin. Scale bar: 200 µm. The results correspond to three independent experiments. (**B**) CPP internalization in C57BL/6 N mice in normal and hyperpolarized conditions. Mice were injected with 5 µM FITC-TAT-RasGAP$_{317-326}$(W317A) with or without 10 µM valinomycin (n = 11 injections per condition). In both panels, the p-values associated with the comparisons of the 'CPP' and 'CPP + valinomycin' conditions were calculated using two-tailed paired t-tests.

The online version of this article includes the following figure supplement(s) for figure 6:

**Figure supplement 1.** Zebrafish and mouse membrane potential modulation.

that they use for their own entry, the formation of inverted micelles in the plasma membrane that translocate the CPPs, or diffusion of the CPPs across the plasma membrane (*Bechara and Sagan, 2013*; *Futaki et al., 2013*; *Guidotti et al., 2017*; *Koren and Torchilin, 2012*; *Trabulo et al., 2010*). Our simulation and cellular data, while providing no evidence for such models, demonstrate that CPP cellular internalization is potassium channel- and V$_m$-dependent in vitro and in vivo. Potassium channels are required to establish a basal low V$_m$, subsequently permissive for CPP direct translocation. Hyperpolarizing drugs, such as valinomycin, enhance permissiveness. When CPPs come into contact with the plasma membrane, they decrease even more the V$_m$, resulting in a locally megapolarized membrane. This increases the likelihood of water pore formation that the CPPs then use to penetrate into cells according to their electrochemical gradient (*Figure 7*). Water pores are created by a combination of lipid head group reorientation coupled to intrusion of a column of water in the membrane bilayer. Water movement plays therefore an active role in the formation of the pore and is not merely occurring once the pores are formed. The movement of the positive charges carried by the CPPs into the cell, as well as the transport of extracellular cations (e.g. Na$^+$), dissipates the V$_m$, resulting in the collapse of the water pores and sealing of the plasma membrane. CPP-mediated formation of water pores is therefore transient and does not affect cell viability. Multiple rounds of CPP-driven water pore formation and CPP translocation into cells can lead to intracellular accumulation of the CPP to concentrations higher than found outside cells (*Figure 4—figure supplement 1C*).

It has not been possible to measure directly the precise values of the V$_m$ that allow the formation of water pores used by CPPs to enter cells. Using an indirect calculation mode based on the uptake of PI alongside CPPs, we have estimated that a V$_m$ in the order of –150 mV is required for water pores to be formed (*Figure 5F*). This might be an underestimation however as modeling data indicate that,

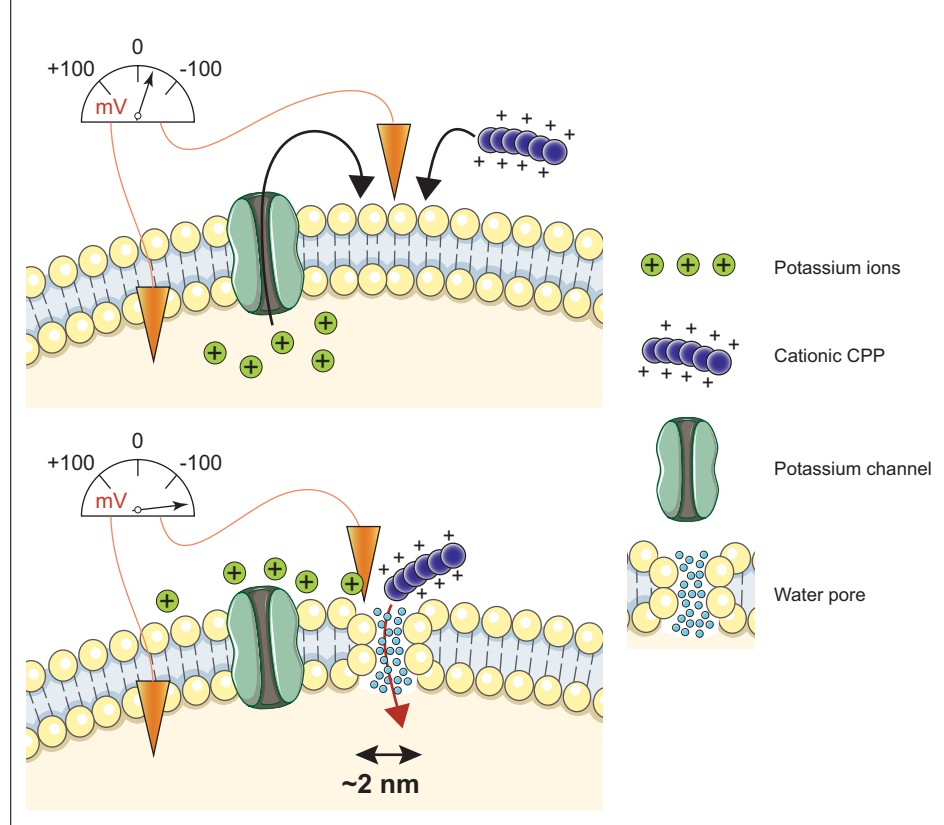

**Figure 7.** Model of cell-penetrating peptide (CPP) direct translocation through water pores. Cationic CPP translocation across cellular membranes is favored by the opening of potassium channels or by hyperpolarizing drugs, such as valinomycin. This sets a sufficiently low membrane potential permissive for CPP direct translocation. When cationic CPPs bind to these already polarized membranes, they induce megapolarization (i.e. a membrane potential estimated to be −150 mV or lower). This leads to the formation of water pores that are then used by CPPs to enter cells.

at −150 mV, the free energy barrier, while being markedly diminished compared to those calculated at −80 or 0 mV, is not fully abrogated (*Figure 4D*). Possibly therefore, the local $V_m$ where CPPs interact with the plasma membranes is much lower than −150 mV and/or changes in CPP structures occur (e.g. refolding, aggregation) leading to further reduction in free energy barrier.

It is worth mentioning that the applied coarse-grained MARTINI force field, as any other model, has a number of known limitations (*Marrink et al., 2019*; *Marrink and Tieleman, 2013*) such as the chemical and spatial resolution, which are both limited compared to atomistic models. There is also a shifted balance between entropy and enthalpy due to the reduced number of degrees of freedom. Moreover, the secondary structure is an input parameter of the model, which implies that secondary structure elements remain fixed during the simulation (*Monticelli et al., 2008*). However, the coarse-grained approach has provided reliable results in the context of protein-membrane interactions and peptide translocation (*Castillo et al., 2013*; *Koch et al., 2019*; *Marrink et al., 2003*; *Monticelli et al., 2008*; *Monticelli et al., 2010*; *Periole et al., 2009*; *Periole et al., 2007*; *Ramadurai et al., 2010*; *Yesylevskyy et al., 2010*). Moreover, the ability of MARTINI coarse-grained force field to model realistic and heterogeneous membranes has been repeatedly reported in literature, as summarized in a recent review paper (*Marrink et al., 2019*).

Our model posits that the number of positively charged amino acids influence the ability of CPPs to hyperpolarize cells and hence to form water pores that they take to translocate into cells. CPP hydropathy strongly correlates with penetration of water molecules in the lipid bilayer, thus supporting the hypothesis that the amount of water each CPP can route inside the membrane is modulated by the hydrophobic and hydrophilic character of the peptide (*Grasso et al., 2018*). The nature of cationic amino acids in peptides determines their translocation abilities. It is known for example that peptides

made of nine lysines (K9) poorly reaches the cytosol (*Figure 5B* and *Figure 3—figure supplement 2C*) and that replacing arginine by lysine in Penetratin significantly diminishes its internalization (*Amand et al., 2012*; *Mitchell et al., 2000*). According to our model, K9 should induce megapolarization and formation of water pores that should then allow their translocation into cells. However, it has been determined that, once embedded into membranes, lysine residues tend to lose protons (*Armstrong et al., 2016*; *Li et al., 2013*; *MacCallum et al., 2008*). This will thus dissipate the strong membrane potential required for the formation of water pores and prevent lysine-containing CPPs to cross the membrane. In contrast, arginine residues are not deprotonated in membranes and water pores can therefore be maintained allowing the arginine-rich CPPs to be taken up by cells. This phenomenon was not modeled in our coarse-grained in silico simulations because the protonation state was fixed at the beginning of the simulation runs and was not allowed to evolve. An additional potential explanation for the internalization differences observed between arginine- and lysine-rich peptides is that even though both arginine and lysine are basic amino acids, they differ in their ability to form hydrogen bonds, the guanidinium group of arginine being able to form two hydrogen bonds (*Fromm et al., 1995*) while the lysyl group of lysine can only form one. Compared to lysine, arginine would therefore form more stable electrostatic interactions with the plasma membrane. According to previously published studies (*Kosuge et al., 2008*; *Mitchell et al., 2000*), the optimal length of consecutive arginine residues appears to be between 9 and 16 amino acids, resulting in optimal CPP cytosolic acquisition. Shorter and longer peptides have decreased internalization efficiencies. The role of the $V_m$ presented in our model is consistent with the reduced uptake of short polyarginine peptides but the $V_m$ parameter of our model cannot explain why longer polyarginine peptides are less efficiently taken up by cells. Our work however also indicates that the water pores created by megapolarization have a diameter of about 2 (–5) nm. Molecules larger than 2 nm are therefore less efficiently transported through these water pores and if polyarginine peptides reach that size their internalization will be hindered. The efficiency of direct translocation of peptides is therefore likely modulated by their sizes, the secondary structures they adopt, and the number of positive charges they carry.

Cationic residues are not the only determinant in CPP direct translocation. The presence of tryptophan residues also plays important roles in the ability of CPPs to cross cellular membranes. This can be inferred from the observation that Penetratin, despite only bearing three arginine residues, can penetrate cells with similar propensities compared to R9 or TAT that contain 9 and 8 arginine residues, respectively. The aromatic characteristics of tryptophan is not sufficient to explain how it favors direct translocation as replacing tryptophan with the aromatic amino acid phenylalanine decreases the translocation potency of the RW9 (RRWWRRWRR) CPP (*Derossi et al., 1994*). Rather, differences in the direct translocation promoting activities of tryptophan and phenylalanine residues may come from the higher lipid bilayer insertion capability of tryptophan compared to phenylalanine (*Christiaens et al., 2002*; *Jobin et al., 2015*; *MacCallum et al., 2008*). There is a certain degree of interchangeability between arginine and tryptophan residues as demonstrated by the fact that replacing up to four arginine residues with tryptophan amino acids in the R9 CPP preserves its ability to enter cells (*Walrant et al., 2020*). Therefore, despite the importance of the membrane potential for CPP direct translocation into cells, other factors also appear to play a role in this process.

While the nature of the CPPs likely dictate their uptake efficiency as discussed in the previous paragraph, the composition of the plasma membrane could also modulate how CPPs translocate into cells. In the present work, we have recorded CPP direct translocation in transformed or cancerous cell lines as well as in primary cells. These cells display various abilities to take up CPPs by direct translocation and the present work indicates that this is modulated by their $V_m$. But as cancer cells display abnormal plasma membrane composition (*Szlasa et al., 2020*), it will be of interest in the future to determine how important this is for their capacity to take up CPPs.

We propose, based on the work described here, that hyperpolarization induced by drugs such as valinomycin represents a simple alternative or parallel approach to optimize CPP internalization. However, hyperpolarizing drugs may be toxic when systemically applied. For example, valinomycin at the concentrations used to induce hyperpolarization (10 µM) would be lethal if systemically injected in mice (LD50 in the low micromolar range; *Daoud and Juliano, 1986*). On the other hand, local administration of valinomycin is far less toxic (*Gad et al., 1985*; *Waksman, 1953*) as confirmed

here in zebrafish and mice. Hyperpolarizing agents may therefore be preferentially used for local or topical applications, which is incidentally the case for the clinically approved CPPs (*Abes et al., 2007*; *Abraham et al., 2015*).

Strategies to improve CPP delivery are becoming increasingly elaborate through the use of nanoparticles (*Bansal et al., 2018*), double-coated nanoparticles (*Khalil et al., 2018*), liposome-polycation-DNA complexes (*Wu et al., 2018*), branched peptides (*Jeong et al., 2016*), etc. Our molecular characterization of the process of CPP direct translocation can be taken advantage of to (i) improve or optimize 'old' CPPs, (ii) design new CPPs, (iii) help explain the behavior of newly discovered CPPs (*Du et al., 2011*; *Kauffman et al., 2018*; *Yin et al., 2009*), (iv) discriminate between target cells and cells that should be left unaffected based on $V_m$, and (v) distinguish between direct translocation and endosomal escape. The present work indicates that the impact on megapolarization should be evaluated when chemical modifications are performed on cationic CPPs to augment their delivery capacities.

# Materials and methods

## Key resources table

| Reagent type (species) or resource | Designation | Source or reference | Identifiers | Additional information |
|---|---|---|---|---|
| Antibody | Anti-V5 (rabbit polyclonal) | Bethyl | Cat#A190-A120 | WB (1:1000) |
| Antibody | Anti-FLAG (Mouse monoclonal) | Sigma-Aldrich | Cat#F1804 | WB (1:1000) |
| Antibody | Anti-Actin (Rabbit monoclonal) | Cell Signaling | Cat#4970 | WB (1:1000) |
| Antibody | Anti-$\alpha$-Tubulin (Rat monoclonal) | Santa Cruz | Cat#sc-51715 | WB (1:1000) |
| Chemical compound, drug | Puromycin | Thermo Fisher | Cat#A11138-002 | 10 µg/ml |
| Chemical compound, drug | Blasticidin | Applichem | Cat#A3784 | 10 µg/ml |
| Chemical compound, drug | XE-991 | Alomone Labs | Cat#X-100 | 10 µg/ml |
| Chemical compound, drug | TRAM-34 | Alomone Labs | Cat#T-105 | 10 µg/ml |
| Chemical compound, drug | Hoechst 3342 | Thermo Fisher | Cat#H21492 | 10 µg/ml |
| Chemical compound, drug | Trypan Blue 0.4% | Life Technologies | Cat#1520061 | |
| Chemical compound, drug | AlexaFluor488-Transferrin | Thermo Fisher | Cat#13342 | 20 µg/ml |
| Chemical compound, drug | TexasRed-Dextran 3000 | Thermo Fisher | Cat#D3329 | 200 µg/ml |
| Chemical compound, drug | TMR-Dextran 10000 | Thermo Fisher | Cat#D1816 | 200 µg/ml |

*Continued on next page*

*Continued*

| Reagent type (species) or resource | Designation | Source or reference | Identifiers | Additional information |
|---|---|---|---|---|
| Chemical compound, drug | TexasRed-Dextran 40000 | Thermo Fisher | Cat#D1829 | 200 µg/ml |
| Chemical compound, drug | Valinomycin | Sigma-Aldrich | Cat#V0627 | 10 µM |
| Chemical compound, drug | Gamicidin | Sigma-Aldrich | Cat#G5002 | 2 µg/ml |
| Chemical compound, drug | Tetraethylammonium | Sigma-Aldrich | Cat#T2265 | 5 mM |
| Chemical compound, drug | Propidium Iodide | Sigma-Aldrich | Cat#81845 | 32 µg/ml PI |
| Chemical compound, drug | DiBac4(3) | Thermo Fisher | Cat#B438 | |
| Chemical compound, drug | Saponin | Sigma-Aldrich | Cat#4706 | 0.1% |
| Chemical compound, drug | Restriction enzyme: BamHI | New England Biolabs | Cat#R313614 | |
| Chemical compound, drug | Restriction enzyme: XmaI | New England Biolabs | Cat#0180 S | |
| Chemical compound, drug | Restriction enzyme: XhoI | New England Biolabs | Cat#R0146L | |
| Chemical compound, drug | Restriction enzyme: HindIII | Promega | Cat#R6041 | |
| Peptide, recombinant protein | TAT-RasGAP317-326 | Biochemistry Department, University of Lausanne | N/A | |
| Peptide, recombinant protein | TAT-RasGAP317-326 | SBS Genetech | N/A | |
| Peptide, recombinant protein | TAT-RasGAP317-326 | Creative Peptides | N/A | |
| Peptide, recombinant protein | FITC-TAT-RasGAP317-326 | Biochemistry Department, University of Lausanne | N/A | |
| Peptide, recombinant protein | FITC-TAT-RasGAP317-326 | Creative Peptides | N/A | |
| Peptide, recombinant protein | FITC-TAT-RasGAP317-326 | SBS Genetech | N/A | |

*Continued on next page*

*Continued*

| Reagent type (species) or resource | Designation | Source or reference | Identifiers | Additional information |
|---|---|---|---|---|
| Peptide, recombinant protein | TMR-TAT-RasGAP317-326 | Creative Peptides | N/A | |
| Peptide, recombinant protein | FITC-TAT-RasGAP317-326(W317A) | Creative Peptides | N/A | |
| Peptide, recombinant protein | FITC-TAT | SBS Genetech | N/A | |
| Peptide, recombinant protein | TAT | SBS Genetech | N/A | |
| Peptide, recombinant protein | FITC-R9 | Biochemistry Department, University of Lausanne | N/A | |
| Peptide, recombinant protein | FITC-Penetratin | Biochemistry Department, University of Lausanne | N/A | |
| Peptide, recombinant protein | FITC-MAP | Biochemistry Department, University of Lausanne | N/A | |
| Peptide, recombinant protein | FITC-Transportan | Biochemistry Department, University of Lausanne | N/A | |
| Peptide, recombinant protein | FITC-D-JNKI1 | SBS Genetech | N/A | |
| Peptide, recombinant protein | FITC-K9 | SBS Genetech | N/A | |
| Commercial assay or kit | TA cloning | Thermo Fisher | Cat#K202020 | |
| Commercial assay or kit | QuikChange II XL Site-Directed Mutagenesis Kit | Aligent | Cat#200522 | |
| Commercial assay or kit | Dual-Luciferase Reporter Assay | Promega | Cat#E1910 | |
| Cell line (*Homo sapiens*) | Raji | Laboratory of Aimable Nahimana | CCL-86 (ATCC) | |
| Cell line (*Homo sapiens*) | SKW6.4 | Laboratory of Pascal Schneider | TIB-215 (ATCC) | |
| Cell line (*Homo sapiens*) | HeLa | ATCC | CCL-2 | |
| Strain, strain background (*Mus musculus*) | C57BL/6NCrl | Charles River Laboratories | C57BL/6NCrl | |

*Continued on next page*

*Continued*

| Reagent type (species) or resource | Designation | Source or reference | Identifiers | Additional information |
|---|---|---|---|---|
| Strain, strain background (Sprague-Dawley *rat*) | Sprague-Dawley | Janvier Laboratories | Sprague-Dawley | |
| Experimental Models (*Danio rerio*) | AB line | European Zebrafish Resource Center | Cat#1175 | |
| Sequence-based reagent | Primer: PCR amplification of FLAG-hKCNQ5 from pShuttlw-FLAG-hKCNQ5(G278S)-IRES-hrGFP2 Forward: | This paper | PCR primers | CATCGGGATCCGCTATACCGGCCACCATGGATTACAAGGA |
| Sequence-based reagent | Primer: PCR amplification of FLAG-hKCNQ5 from pShuttlw-FLAG-hKCNQ5(G278S)-IRES-hrGFP2 Reverse: | This paper | PCR primers | CATCGCCCGGGGCTATACCGTACCGTCGACTGCAGAATTC |
| Sequence-based reagent | Primer: introducing silent mutations in FLAG-hKCNQ5(G278S)-IRES-Neo Forward: | This paper | PCR primers | AAA TAA GAA CCA AAA ATC CTA TGT ACC ATG CCG TTA TCA GCT CCT TGC TGT GAG CAT AAA CCA CTG AAC CCA G |
| Sequence-based reagent | Primer: introducing silent mutations in FLAG-hKCNQ5(G278S)-IRES-Neo Reverse: | This paper | PCR primers | CTG GGT TCA GTG GTT TAT GCT CAC AGC AAG GAG CTG ATA ACG GCA TGG TAC ATA GGA TTT TTG GTT CTT ATT T |
| Sequence-based reagent | Primer: reverting G278S mutation in FLAG-hKCNQ5(SM, G278S)-IRES-Neo Forward: | This paper | PCR primers | TTT TGT CTC CAT AGC CAA TAG TTG TCA ATG TAA TTG TGC CCC |
| Sequence-based reagent | Primer: reverting G278S mutation in FLAG-hKCNQ5(SM, G278S)-IRES-Neo Reverse: | This paper | PCR primers | GGG GCA CAA TTA CAT TGA CAA CTA TTG GCT ATG GAG ACA AAA |
| Sequence-based reagent | sgRNA targeting KCNQ5, KCNN4 and KCNK5, see *Supplementary file 5* | This paper | PCR primers | |
| Sequence-based reagent | Primer: first PCR to amplify the lentiCRISPR sgRNA region Forward: | *Shalem et al., 2014* | PCR primers | AATGGACTATCATATGCTTACCGTAACTTGAAAGTATTTCG |
| Sequence-based reagent | Primer: first PCR to amplify the lentiCRISPR sgRNA region Reverse: | *Shalem et al., 2014* | PCR primers | CTTTAGTTTGTATGTCTGTTGCTATTATGTCTACTATTCTTTCC |
| Sequence-based reagent | Primers used during the second PCR to attach Illumina adaptors with barcodes, see *Supplementary file 4* | *Shalem et al., 2014* | PCR primers | |
| Recombinant DNA reagent | Plasmid:hKCNN4-V5.lti | DNASU | HsCD00441560 | |
| Recombinant DNA reagent | Plasmid: hKCNK5-FLAG.dn3 | GenScript | OHu13506 | |
| Recombinant DNA reagent | Plasmid: Myc-mKCNJ2-T2A-IRES-tdTomato.lti | *Xue et al., 2014* | Addgene Plasmid #60598 | |

*Continued on next page*

*Continued*

| Reagent type (species) or resource | Designation | Source or reference | Identifiers | Additional information |
|---|---|---|---|---|
| Recombinant DNA reagent | Plasmid: LeGo-iT2 | *Weber et al., 2008* | Addgene Plasmid #27343 | |
| Recombinant DNA reagent | Plasmid: pMD2.G | Didier Trono Laboratory | Addgene Plasmid #12259 | |
| Recombinant DNA reagent | Plasmid: psPAX2 | Didier Trono Laboratory | Addgene Plasmid #12260 | |
| Recombinant DNA reagent | Plasmid: FLAG-hKCNQ5(G278S)-IHRES-NeoR | This paper | | |
| Recombinant DNA reagent | Plasmid: pShuttle-FLAG-hKCNQ5(G278S)-IRES-hrGFP2 | Kenneth L Byron Laboratory | N/A | |
| Recombinant DNA reagent | Plasmid: TRIP-PGK-IRES-Neo | Didier Trono Laboratory | N/A | |
| Recombinant DNA reagent | Plasmid: FLAG-hKCNQ5(SM,G278S)-IRES-Neo | This paper | | |
| Recombinant DNA reagent | Plasmid: FLAG-hKCNQ5(SM)-IRES-Neo | This paper | | |
| Recombinant DNA reagent | Plasmid:pLUC705 | Bin Yang Laboratory | N/A | |
| Recombinant DNA reagent | Plasmid: LeGo-iG2 | *Weber et al., 2008* | Addgene Plasmid #27341 | |
| Recombinant DNA reagent | Plasmid: pLUC705.LeGo-iG2 | This paper | | |
| Recombinant DNA reagent | Plasmid: pTAT-Cre | *Wadia et al., 2004* | Addgene Plasmid #35619 | |
| Recombinant DNA reagent | Plasmid: Cre-reporter.lti | *D'Astolfo et al., 2015* | Addgene Plasmid #62732 | |
| Recombinant DNA reagent | Plasmid: GeCKO v2 library | *Shalem et al., 2014* | Addgene Plasmid #1000000049 | |
| Software, algorithm | ImageJ | *Schneider et al., 2012* | https://imagej.nih.gov/ij/ | |
| Software, algorithm | Zeiss Zen Lite 2.3 | Carl Zeiss Microscopy GmbH | https://www.zeiss.fr/microscopie/produits/microscope-software/zen-lite.html | |
| Software, algorithm | MultiClamp 2.2.0 | Axon MultiClamp (Molecular Devices) | http://mdc.custhelp.com/app/answers/detail/a_id/18877/~/axon%E2%84%A2-multiclamp%E2%84%A2-commander-software-download-page | |

*Continued on next page*

*Continued*

| Reagent type (species) or resource | Designation | Source or reference | Identifiers | Additional information |
|---|---|---|---|---|
| Software, algorithm | Clampfit 10.7.0 | Axon pCLAMP (Molecular Devices) | http://mdc.custhelp.com/app/answers/detail/a_id/18779/~/axon%E2%84%A2pclamp%E2%84%A2-10-electrophysiology-data-acquisition-%26-analysis-software-download | |
| Software, algorithm | Kaluza 1.3 | Beckman Coulter | https://www.beckman.ch/flow-cytometry/software/kaluza | |
| Software, algorithm | Gen5.2.5 | BioTek Instruments | https://www.biotek.com/products/software-robotics-software/gen5-microplate-reader-and-imager-software/ | |
| Software, algorithm | GloMax | Promega | https://ch.promega.com/resources/software-firmware/detection-instruments-software/promega-branded-instruments/glomax-96-microplate-luminometer/ | |
| Software, algorithm | GraphPad Prism8 | GraphPad | https://www.graphpad.com/scientific-software/prism/ | |
| Software, algorithm | MicroCal ITC200 | Malvern Panalytical | N/A | |
| Software, algorithm | Clone Manager9 | Sci-Ed Software | https://www.scied.com/dl_cm10.htm | |
| Software, algorithm | Li-Cor Odyssey | LI-COR Biosciences | N/A | |
| Software, algorithm | GROMACS 2018.3 | http://www.gromacs.org/Downloads | N/A | |
| Software, algorithm | VISUAL MOLECULAR DYNAMICS (VMD) | https://www.ks.uiuc.edu/Development/Download/download.cgi?PackageName=VMD | N/A | |
| Software, algorithm | XMGRACE 5.1 | http://plasma-gate.weizmann.ac.il/Grace/ | N/A | |

*Continued on next page*

*Continued*

| Reagent type (species) or resource | Designation | Source or reference | Identifiers | Additional information |
|---|---|---|---|---|
| Software, algorithm | PEP-FOLD SERVER | http://mobyle.rpbs.univ-paris-diderot.fr/cgi-bin/portal.py#forms::PEP-FOLD | N/A | |
| Other | RPMI-like media without KCl and NaCl | Biowest | N/A | |

## Chemicals

Puromycin 10 mg/ml (Thermo Fisher, ref no. A11138-02) was aliquoted and stored at –20 °C. Blasticidin (Applichem, ref no. A3784) was dissolved at 1 mg/ml in water and stored at –20 °C. XE-991 and TRAM-34 (Alomone Labs, ref no. X-100 and T-105, respectively) was dissolved in DMSO at 100 mM and stored at –20 °C. Cells were pre-incubated with 10 µM of these inhibitors for 30 min and then kept throughout the experiments. Live Hoechst 33342 (Sigma, ref no. CDS023389) was aliquoted and stored at –20 °C. Trypan Blue 0.4 % (Life Technologies, ref no. 15250061) was stored at room temperature. AlexaFluor488-labeled human transferrin was dissolved in PBS at 5 mg/ml and stored at 4 °C (Thermo Fisher, ref no. 13342). TexasRed-labeled neutral 3000 and 40,000 Da dextran was dissolved in PBS at 10 mg/ml and stored at –20 °C (Thermo Fisher, ref no. D3329 and D1829, respectively). TMR-labeled 10,000 neutral dextran was dissolved in PBS at 10 mg/ml and stored at –20 °C (Thermo Fisher, ref no. D1816).

## Antibodies

The rabbit polyclonal anti-V5 (Bethyl, ref no. A190-A120), mouse monoclonal anti-FLAG antibody was from Sigma-Aldrich (ref no. F1804), rabbit monoclonal anti-actin (Cell Signaling, ref no. 4970), and rat monoclonal anti-γ-tubulin (Santa Cruz, ref no. sc-51715) antibodies were used for Western blotting.

## Cell lines

All cell lines were culture in 5 % $CO_2$ at 37 °C. Raji (kind gift from the laboratory of Aimable Nahimana, ATCC: CCL-86), SKW6.4 (kind gift from the laboratory of Pascal Schneider, ATCC: TIB-215), and HeLa (ATCC: CCL-2) cells were cultured in RPMI (Invitrogen, ref no. 61870) supplemented with 10 % heat-inactivated FBS (Invitrogen, ref no. 10270–106). HEK293T cells (ATCC: CRL-3216) were cultured in DMEM supplemented with 10 % FBS and were used here only for lentiviral production. All cell lines were mycoplasma-negative and authenticated via Microsynth cell authentication service. Unless, otherwise indicated, experiments were performed in RPMI with 10 % FBS.

## Zebrafish

Zebrafish (*Danio rerio*) from AB line were bred and maintained in our animal facility under standard conditions (*Marrink et al., 2019*), more specifically at 28.5 °C and on a 14:10  hr light:dark cycle at the Zebrafish facility of the Faculty of Biology and Medicine, University of Lausanne (cantonal veterinary approval VD-H21). Zebrafish of 20  hr post fertilization were collected and treated with 0.2 mM phenylthiourea (Sigma, St Louis, MO) to suppress pigmentation. Embryos were raised at 28.5 °C in Egg water (0.3 g sea salt/l reverse osmosis water) up to 4 days post fertilization.

## Mice

C57BL/6NCrl were acquired from Charles River laboratories, which were then housed and bred in our animal facility. All experiments were performed according to the principles of laboratory animal care and Swiss legislation under ethical approval (Swiss Animal Protection Ordinance; permit number VD3374.a).

## Primary cortical neuronal culture

Sprague-Dawley rat pups (from Janvier, France) were euthanized in accordance with the Swiss Laws for the protection of animals, and the procedures were approved by the Vaud Cantonal Veterinary

Office (permit number VD1407.9). Primary neuronal cultures from cortices of 2-day-old rats were prepared and maintained at 37 °C with a 5 % $CO_2$-containing atmosphere in neurobasal medium (Life Technologies, 21103–049) supplemented with 2 % B27 (Invitrogen, 17504044), 0.5 mM glutamine (Sigma, G7513), and 100 µg/ml penicillin-streptomycin (Invitrogen, 15140122) as described previously (*Vaslin et al., 2007*). Neurons were plated at a density of ~3 × $10^5$ cells on 12 mm glass coverslips coated with 0.01 % poly-L-lysine (Sigma, P4832). Half of the medium was changed every 3–4 days and experiments were performed at 12–13 days in vitro.

## Confocal microscopy

Confocal microscopy experiments were done on live 300,000 cells. Cells were seeded for 16 hr onto glass bottom culture dishes (MatTek, corporation ref no. P35G-1.5–14 C) in 2 ml RPMI, 10 % FBS and treated as described in the figures in 1 ml media, 10% FBS. For nuclear staining, 10 µg/ml live Hoechst 33342 (Molecular Probes, ref no. H21492) was added in the culture medium 5 min before washing cells twice with PBS. After washing, cells were examined with a plan Apochromat 63 × oil immersion objective mounted on a Zeiss LSM 780 laser scanning fluorescence confocal microscope equipped with gallium arsenide phosphide detectors and three lasers (a 405 nm diode laser, a 458-476-488-514 nm argon laser, and a 561 nm diode-pumped solid-state laser). Time-lapse experiments were done using an incubation chamber set at 37 °C, 5 % $CO_2$ and visualized with a Zeiss LSM710 Quasar laser scanning fluorescence confocal microscope equipped with either Neofluar 63 ×, 1.2 numerical aperture (NA) or plan Neofluar 100 ×, 1.3 NA plan oil immersion objective (and the same lasers as above). Visual segregation of cells based on types of CPP entry, associated with either vesicular or diffuse cytosolic staining, was performed as shown in *Figure 1—figure supplement 1A*. Cell images were acquired at a focal plane near the middle of the cell making sure that nuclei were visible. Image acquisition was performed using identical settings for the data presented in a given panel and the related supplementary information.

## Flow cytometry

Flow cytometry experiments were performed using a Beckman Coulter FC500 instrument. Cells were centrifuged and resuspended in PBS prior to flow cytometry. Data analysis was done with Kaluza Version 1.3 software (Beckman Coulter).

## Cell death and CPP internalization measurements

With the exception of neurons, cell death was quantitated with 8 µg/ml PI (Sigma, ref no. 81845). Unless otherwise indicated, cell death was assessed after 16 hr of continuous incubation in Raji and SKW6.4 cells and 24 hr in HeLa cells. Prior to treatment, 300,000 cells were seeded in six-well plates for 16 hr in 2 ml media, 10 % FBS. Treatment was done in 1 ml media with 10% FBS. Cell death and peptide internalization were analyzed by flow cytometry. Internalization measurements were done after 1 hr of incubation. Peptide internalization in primary cortical neurons was assessed by confocal microscopy. Cell-associated fluorescence was quantitated with ImageJ. When cytosolic fluorescent was recorded with ImageJ, the regions of interest that were analyzed were chosen so as not to contain labeled endosomes (*Figure 1—figure supplement 2D*, circle).

## Lentivirus production

Recombinant lentiviruses were produced as described (*Marrink et al., 2019*; *Melikov et al., 2015*) with the following modification: the envelope plasmid pMD.G and the packaging vector pCMVΔR8.91 were replaced by pMD2.G and psPAX2, respectively.

## In vitro membrane potential measurements

Two methods were used to assess cellular membrane potential in vitro. With the first method, the membrane potential was determined by incubating 300,000 cells for 40 min with 100 nM of the fluorescent probe DiBAC4(3) (Thermo Fisher, ref no. B438) in six-well plates in 1 ml media, 10 % FBS, and the median fluorescence intensity was then assessed by flow cytometry. Calculation of the actual membrane potential in mV based on the DiBAC4(3) signals was performed as described earlier (*Klapperstück et al., 2009*; *Krasznai et al., 1995*). The second method relied on electrophysiology recordings. To perform these, the bath solution composition was (in mM): 103.9 NaCl, 23.9 $NaHCO_3$,

2 CaCl$_2$, 1.2 MgCl$_2$, 5.2 KCl, 1.2 NaH$_2$PO$_4$, 2 glucose, and 1.7 ascorbic acid. The pipet solution was composed of (in mM): 140 KMeSO$_4$, 10 HEPES, 10 KCl, 0.1 EGTA, 10 phosphocreatine, and 4 MgATP. The patch pipets had a resistance of 2.4–3.6 M$\Omega$. Perforated patch recordings were performed as previously described (*Cueni et al., 2008*). Briefly, freshly prepared gramicidin D (Sigma, ref no. G5002), at 2.8 µM final concentration, was added to prefiltered patch pipet solution and then sonicated for three consecutive times during 10 s. Cell-attached configuration was achieved by applying negative pressure on patch pipet until seal resistance of over 1 G$\Omega$ was reached. After gaining cell access through gramicidin created pores, membrane potential measurements were done in current clamp at 0 pA for at least 3 min. Since primary rat neurons are killed following full depolarization induced by gramicidin, the standard curve from membrane potential calculations (*Klapperstück et al., 2009*; *Krasznai et al., 1995*) was performed using gramicidin-treated Raji cells incubated with increasing concentrations of DiBac4(3) for 40 min. Images of the cells were then taken using an LSM780 confocal microscope and the cell-associated fluorescence quantitated with ImageJ.

## Relative membrane potential assessment in vivo

Zebrafish embryos in Egg water (see 'Zebrafish' section) were incubated for 40 min in the presence or in the absence of various concentrations of valinomycin together with 950 nM DiBac4(3). The embryos were then fixed and visualized under a confocal LSM710 microscope (*Adams and Levin, 2012*). DiBac4(3)-associated fluorescence of a region of interest (ROI) of about 0.0125 mm$^2$ in the tail region was quantitated with ImageJ. The values were normalized to the control condition (i.e. in the absence of valinomycin). Mice were intradermally injected with 10 µl of a 950 nM DiBac4(3) PBS solution containing or not 10 µM valinomycin and sacrificed 1 hr later. The skin was excised, fixed in 4 % formalin, paraffin-embedded, and used to prepare serial histological slices. Pictures of the slices were taken with a CYTATION3 apparatus. The DiBac4(3)-associated fluorescence in the whole slice was quantitated with ImageJ. The slice in the series of slices prepared from a given skin sample displaying the highest fluorescence signal was considered as the one nearest to the injection site. The signals from such slices are those reported in the figures.

## Experimental modulation of the plasma membrane potential

### Flow cytometry assessment of CPP internalization

Raji, SKW6.4, or HeLa cells: 300,000 cells were plated on non-coated plates to avoid cell adherence in 500 µl RPMI, 10 % FBS. Cellular depolarization was induced by pre-incubating the cells at 37 °C with 2 µg/ml gramicidin for 5 min and/or by placing them in potassium-rich buffer (*Hirose et al., 2012*) for 30 min (40 mM KCl, 100 mM potassium glutamate, 1 mM MgCl$_2$, 1 mM CaCl$_2$, 5 mM glucose, 20 mM HEPES, pH 7.4). Cells were then treated with the selected CPPs at the indicated concentrations for 1 hr when peptide internalization was recorded or with 100 nM DiBac4(3) for 40 min when membrane potential needed to be measured. Hyperpolarization in Raji cells in the presence of TAT-RasGAP$_{317-326}$ was performed by treating the cells with 10 µM valinomycin for 20 min in RPMI without serum. Cells were then treated with 5 µM TAT-RasGAP$_{317-326}$ for 1 hr or 100 nM DiBac4(3) for 40 min. In the case of SKW6.4 and HeLa cells, hyperpolarization was induced by infection with a viral construct expressing KCNJ2 (see 'Lentivirus production' section). Cells were then treated with 40 µM of indicated CPP for 1 hr or 100 nM DiBac4(3) for 40 min.

### CPP cytosolic internalization quantitation based on confocal microscopy

Three-hundred thousand wild-type HeLa cells were plated overnight on glass-bottom dishes in 2 ml RPMI with 10 % FBS. The next day, serum was removed and cells were pre-incubated at 37 °C with 2 µg/ml gramicidin for 5 min, 10 µM valinomycin for 20 min, or were left untreated in 1 ml media with 10% FBS. The indicated CPPs were then added and cells were incubated for 1 hr at 37 °C. Cells were then washed and visualized in RPMI without serum under a confocal microscope. CPP cytosolic internalization was quantitated within a cytosolic region devoid of endosomes using ImageJ. The number of CPP-positive vesicles per cell was visually determined in a given focal plane.

Neurons (12 days post isolation) were pre-incubated 30 min with 5 mM TEA (tetraethylammonium, Sigma Aldrich, ref no. T2265; gramicidin is toxic in these neurons; see section 'In vitro membrane potential measurements') to induce depolarization or 10 µM valinomycin to induce hyperpolarization in bicarbonate-buffered saline solution (116 mM NaCl, 5.4 mM KCl, 0.8 mM MgSO$_4$, 1 mM NaH$_2$PO$_4$,

26.2 mM NaHCO$_3$, 0.01 mM glycine, 1.8 mM CaCl$_2$, 4.5 mg/ml glucose) in a 37 °C, 5 % CO$_2$ incubator. The cells were then incubated 1 hr with 2 µM of FITC-labeled TAT-RasGAP$_{317-326}$. The cells were finally washed thrice with PBS and images were acquired using a LSM780 confocal microscope. Cell-associated peptide fluorescence was quantitated using ImageJ.

## Setting membrane potential by changing potassium concentrations in the media

RPMI-like media made without potassium chloride and without sodium chloride was from Biowest (*Supplementary file 1*). Varying concentrations of potassium chloride were added to this medium containing 10 % FBS. Sodium chloride was also added so that the sum of potassium and sodium chloride equaled 119 mM (also taking into account the concentrations of sodium and potassium in FBS). Three-hundred thousand cells were pre-incubated in 1 ml media, 10 % FBS containing different concentrations of potassium for 20 min, then different CPPs at a 40 µM concentration were added and cells were incubated for 1 hr at 37 °C in 5 % CO$_2$. Cells were washed once in PBS and CPP internalization was measured by flow cytometry. The corresponding membrane potential was measured with DiBac4(3).

## In silico CPP translocation free energy assessment through MARTINI coarse-grained simulations

An asymmetric multi-component membrane was constructed and solvated using CHARMM-GUI (*Jo et al., 2008*; *Qi et al., 2015*). Each layer contained 100 lipids (*Supplementary file 2*), in a previously described composition (*Ingólfsson et al., 2014*). The membrane was solvated with 2700 water molecules, obtaining a molecular system of 10,200 particles. The MARTINI force field 2.2 p (*Marrink et al., 2003*; *Wassenaar et al., 2015*) was used to define phospholipids' topology through a coarse-grained approach. The polarizable water model has been used to assess the water topology (*Yesylevskyy et al., 2010*). Each peptide (R9, TAT, and TAT-RasGAP$_{317-326}$) structural model has been obtained by PEPFOLD-3 server (*Lamiable et al., 2016*), as done in previous studies in the field (*Grasso et al., 2015*; *Grasso et al., 2018*; *Serulla et al., 2020*). For each molecular system, one CPP was positioned 3 nm away from the membrane outer leaflet, in the water environment corresponding to the extracellular space. Then, the system was equilibrated through four MD simulations of 100 ps, 200 ps, 500 ps, and 100 ns under the NPT ensemble. Position restraints were applied during the first three MD simulations and gradually removed, from 200 to 10 kJ/mol*nm$^2$. Velocity rescaling (*Bussi et al., 2007*), temperature coupling algorithm, and time constant of 1.0 ps were applied to keep the temperature at 310.00 K. Berendsen (*Berendsen et al., 1984*) semi-isotropic pressure coupling algorithm with reference pressure equal to 1 bar and time constant 5.0 ps was employed. Then, all systems were simulated for the production run in the NPT ensemble with a time step of 20 fs. Electrostatic interactions were calculated by applying the particle-mesh Ewald (*Darden et al., 1993*) method and van der Waals interactions were defined within a cut-off of 1.2 nm. Periodic boundary conditions were applied in all directions. Trajectories were collected every 10 ps and the Visual Molecular Dynamics (VMD) (*Humphrey et al., 1996*) package was employed to visually inspect the simulated systems. Three different transmembrane potential values were considered: 0 , 80 , and 150 mV. In the MD simulations, an external electric field E$_{ext}$ was applied parallel to the membrane normal z, that is, perpendicular to the bilayer surface. This was achieved by including additional forces Fi = q*E$_{ext}$ acting on all charged particles i. In order to determine the effective electric field in simulations, we applied a computational procedure reported in literature (*Gumbart et al., 2012*). A well-tempered metadynamics protocol (*Barducci et al., 2008*) was applied to estimate the free energy landscape of CPP translocation. Two collective variables were considered: the lipid/water density index and the CPP-membrane distance. Gaussian deposition rate of 2.4 kJ/mol every 5 ps was initially applied and gradually decreased on the basis of an adaptive scheme. Gaussian widths of 0.5 and 0.2 nm were applied following a well-established scheme (*Deriu et al., 2016*; *Granata et al., 2013*; *Grasso et al., 2017*; *Laio and Gervasio, 2008*). In particular, the Gaussian width value was of the same order of magnitude as the standard deviation of the distance CV, calculated during unbiased simulations. The well-tempered metadynamics simulations were computed using GROMACS 2019.4 package (*Abraham et al., 2015*) and the PLUMED 2.5 open-source plug-in (*Tribello et al., 2014*). The reconstruction of the free-energy surface was performed by the reweighting algorithm procedure (*Tiwary and Parrinello, 2015*) allowing the

estimation of the free energy landscape. The comparison between the water pore formation free energy estimated by our MARTINI coarse-grained simulations and previous estimations available in literature is reported in *Supplementary file 3*. Each system was simulated (with a 20 fs time step) until convergence was reached. The electrostatic potential maps were computed by the APBS package (*Baker, 2004*) on the molecular system composed of R9 peptide in contact with the cell membrane, without any applied external electrostatic field. In detail, the non-linear Poisson-Boltzmann equation was applied using single Debye-Huckel sphere boundary conditions on a $97 \times 97 \times 127$ grid with a spacing of 1 Å centered at the COM of the molecular system. The relative dielectric constants of the solute and the solvent were set to 2.5 and 78.4, respectively. The ionic strength was set to 150 mM and the temperature was fixed at 310 K (*Baker, 2004*; *Grasso et al., 2017*). The average and standard deviation values of the local transmembrane potential were computed considering 10 different trajectory snapshots taken from the molecular trajectory.

## In silico pore formation kinetics through MARTINI coarse-grained simulations

The same molecular system previously constructed and equilibrated was investigated to estimate the water pore formation kinetics by applying a constant electrostatic potential (*Böckmann et al., 2008*; *Fernández et al., 2010*; *Gao et al., 2019*; *Gumbart et al., 2012*; *Gurtovenko and Lyulina, 2014*; *Kirsch and Böckmann, 2016*; *Tieleman, 2004*; *Ziegler and Vernier, 2008*). Each peptide (R9, TAT, and TAT-RasGAP$_{317-326}$) was positioned 3 nm away from the membrane at the beginning of each simulation. The relatively small size of the molecular system and the application of coarse-grained MARTINI force field allowed us to study the pore formation kinetics, requiring many simulations at varying field strengths. In detail, 25 simulations were performed for each molecular system at different external electric field strengths from 0.0055 to 0.090 V/nm. In the MD simulations, an external electric field $E_{ext}$ was applied parallel to the membrane normal z, that is, perpendicular to the bilayer surface. This was achieved by including additional forces $F_i = q*E_{ext}$ acting on all charged particles i. In order to determine the effective electric field in simulations, we applied a computational procedure reported in literature (*Gumbart et al., 2012*). The results are reported in *Figure 4—figure supplement 1*.

## In silico cell membrane hyperpolarization modeling through ion imbalance in MARTINI coarse-grained simulations

The translocation mechanism of each CPP was studied by ion imbalance in a double-bilayer system (*Delemotte et al., 2008*; *Gao et al., 2019*; *Gurtovenko and Vattulainen, 2007*; *Herrera and Pantano, 2009*). The same asymmetric membrane considered to perform the single-bilayer simulations (*Supplementary file 2*) was used to build up the double-bilayer system. The double-membrane system was solvated with 4300 water molecules, obtaining a molecular system of 20,000 particles. The MARTINI force field 2.2 p (*Marrink et al., 2003*; *Wassenaar et al., 2015*) was used to define phospholipids' topology through a coarse-grained approach. The polarizable water model was used to model the water topology (*Yesylevskyy et al., 2010*). The elastic network ELNEDYN (*Periole et al., 2009*) was applied to reproduce the structural and dynamic properties of the CPPs.

For each molecular system, one CPP was positioned in the middle of the double-bilayer system, 2 nm away from the membrane outer leaflets, in the water environment corresponding to the extracellular space. Then, the system was equilibrated through four MD simulations of 100 ps, 200 ps, 500 ps, and 100 ns under the NPT ensemble. Position restraints were applied during the first three MD simulations and gradually removed, from 200 to 10 kJ/mol*nm$^2$. Velocity rescaling (*Bussi et al., 2007*), temperature coupling algorithm, and time constant of 1 ps were applied to keep the temperature at 310 K. Berendsen (*Berendsen et al., 1984*) semi-isotropic pressure coupling algorithm with reference pressure equal to 1 bar and time constant 5 ps was employed. Then, all systems were simulated for the production run in the NPT ensemble with the time step of 20 fs with Parrinello-Rahman pressure coupling (*Parrinello and Rahman, 1981*).

Membrane hyperpolarization was achieved through a net charge difference of 30 positive ions between intracellular and extracellular space, considering all charged ions of the system and fulfilling the full system electroneutrality. Ten different replicas of each molecular simulation were performed until the water pore formation and closure events were observed. The visual inspection of the simulated molecular systems is reported in *Figure 4—figure supplement 1A*. To analyze whether the

CPPs were able to cross the membrane and reach the intracellular compartment, their trajectories were studied in the last 5 ns of each simulation replica. Considering the CPP position with respect to the membrane bilayers and the CPP's solvent accessible surface area, three different compartments were defined: intracellular space, lipid bilayer (cell membrane), and extracellular space. The radius of the water pores within the membrane was calculated as previously done (*Gurtovenko and Vattulainen, 2007*; *Leontiadou et al., 2004*). We assumed that the central part of the cylindrical water pore contains N water molecules at the same density as outside of the water flux.

## In vitro assessment of water pores

Three-hundred thousand wild-type HeLa cells or primary rat cortical neurons were incubated with 32 µg/ml PI (0.8–1.5 nm diameter 75) or 200 µg/ml dextran of different molecular weight in the presence or in the absence of the indicated CPPs in normal, depolarizing (2 µg/ml gramicidin), or hyperpolarizing (10 µM valinomycin) conditions in 1 ml media, 10 % FBS. Time-lapse images were acquired by confocal microscopy every 10 s. The percentage of cells where direct CPP translocation occurred, as well as the percentage of cells positively stained for PI, were manually quantitated using ImageJ based on snap shot images taken after 30 min of incubation, as shown in *Figure 5—figure supplement 1C*. Cytosolic PI fluorescence was assessed with ImageJ, by selecting a region within the cell's cytosol devoid of endosomes. Cells were permeabilized with 0.1% saponin (Sigma, ref no. 4706, diluted in PBS weight:volume; 30 min incubation at 37 °C in a 5 % $CO_2$ incubator) to determine the maximal PI uptake cell capacity. Three fitting models were obtained:

$$\text{Exponential decline:} y = 2570 \cdot e^{\left(\frac{-x}{54}\right)}$$
$$\text{Exponential:} y = 2570 \cdot e^{-0.02x}$$
$$\text{Modified power:} y = 2570 \cdot 0.98^x$$

These equations fitted equally well the PI uptake/$V_m$ curve in *Figure 5E*. For the calculations used in *Figure 5F*, the exponential decline equation was used.

## Zebrafish viability

FITC-TAT-RasGAP$_{317-326}$(W317A) internalization in zebrafish was assessed either by adding the peptide directly in Egg water or by injection. Experiments in which the peptide was added in the water were performed on fish between 4 and 24 hr post fertilization. Viability assays were done on embryos of 4, 6, and 24 hr post fertilization to determine a maximal nonlethal dose of the peptide that can be used. Different concentrations of the peptide were added to 500 µl water per well in 24-well plate, with between eight and eleven embryos per well. Fish viability was visually assessed at 20 hr post incubation with the peptide. Hyperpolarization-associated viability was visually assessed at different time points in the presence or in the absence of the peptide with or without various valinomycin concentrations. Zebrafish were visualized with binocular microscope and CYTATION3 apparatus. Survival was visually assessed under a binocular microscope by taking into consideration the embryo transparency (as dead embryos appear opaque), general development characteristics, and motility.

## CPP internalization in vivo

To assess peptide internalization in zebrafish, two methods were used: (i) addition of the peptide directly in 500 µl of Egg water in 24-well plates containing between eight and twelve embryos per well and (ii) intramuscular injections. In the case of the first method, after the indicated treatments, zebrafish were washed, fixed in 4 % PFA/PBS for 1 hr at room temperature. Whole embryos were mounted on slides with Fluoromount-G (cBioscience, ref no. 00-4958-02). Zebrafish were then visualized under an LSM710 confocal microscope. Experiments where the peptide was added directly to the water were performed on zebrafish at 18 hr post fertilization to limit cuticle development that would hinder peptide access to the cells. In the case of the second method, 8 nl injections (containing the various combinations of peptide and valinomycin and 0.05 % (vol:vol) phenol red as an injection site labeling agent) were done on 48 hr post fertilization embryos into the tail muscle around the extremity of yolk extension, after chorion removal and anesthesia with 0.02 % (w:vol) tricaine (*Schroeder et al., 2000*) buffered with sodium bicarbonate to pH 7.3. At this age, zebrafish already have well developed tissues that can be easily visually distinguished. Injections were done with an Eppendorf Microinjections FemtoJet 4i apparatus. After the indicated treatments, embryos were fixed in

4 % PFA/PBS and visualized under a confocal microscope. Some embryos were kept alive for viability evaluation post injection until the age of 4 days.

Experiments with mice were performed in 10- to 14 -week-old C57BL/6NCrl mice anesthetized with ketasol/xylasol (9.09 mg/ml ketasol and 1.82 mg/ml xylasol in water; injection: 10 µl per g of body weight). The back of the mice was shaved and intradermic injections were performed (a total of 10 µl was injected). Mice were kept under anesthesia for 1 hr and Artificial tears (Lacryvisc) were used to avoid eye dryness. Mice were then sacrificed by $CO_2$ inhalation, skin was cut at injection sites, fixed in 4 % formalin, and paraffin embedded for histology analysis. For each sample, 10–15 slides were prepared and peptide internalization was visualized with a CYTATION3 apparatus. Fluorescence intensity was quantitated with ImageJ. The slices displaying the highest fluorescence signal were considered as those nearest to the injection site and the fluorescent values from these slides were used in *Figure 6B*.

## Assessment of endosomal escape and direct translocation

Three hundred thousand cells were seeded onto glass-bottom dishes in 2 ml RPMI, 10 % FBS for 16 hr. Quantitation of cytosolic fluorescence was performed within live HeLa cells pre-incubated with 80 µM TAT-RasGAP$_{317-326}$ for 30 min at 37 °C in 1 ml media, with 10 % FBS and then incubated for the indicated periods of time in the presence (i.e. no wash after the pre-incubation) or in the absence (i.e. following three consecutive washes with RPMI supplemented with 10 % FBS) of extracellular labeled peptide. Endosomal escape from lysosomes was induced by 1 mM LLOME (L-leucyl-L-leucine methyl ester) (*Repnik et al., 2017*), added in the 1 ml media, 10 % FBS 30 min after the CPPs were washed out (cells were exposed to LLOME for 100 min). Confocal images were taken every 5 min after the 30 min pre-incubation with the CPPs. For each cell, the fluorescence intensity of one ROI devoid of labeled endosomes throughout the experiment was quantitated over time using ImageJ Time Series Analyzer V3. The surface of the ROI was identical for all cells. Only cells displaying labeled endosomes after the 30 min pre-incubation (that is cells that had taken up the CPPs by endocytosis) were analyzed. Note that the washing steps, for reasons unclear at this time, induced a slightly higher initial ROI intensity signal.

## Transferrin internalization quantitation

Wild-type HeLa cells were plated in 12-well plates (200,000 cells per well) for 16 hr in 1 ml RPMI (Invitrogen, ref no. 61870), supplemented with 10 % heat-inactivated FBS (Invitrogen, ref no. 10270–106). Cells were then incubated with 20 µg/ml AlexaFluor488-conjugated transferrin for 20 min at 37 °C in 5 % $CO_2$. Cells were washed with PBS and pelleted after trypsinization. To quench membrane-bound transferrin fluorescence, cells were resuspended in 0.2 % trypan blue diluted in PBS. Transferrin internalization was quantitated by flow cytometry using Beckman Coulter FC500 instrument. Data analysis was done with Kaluza Version 1.3 software (Beckman Coulter).

## TAT-PNA-induced luciferase activity

The LeGOiG2-LUC705 lentiviral construct (plasmid #975) encodes a luciferase gene interrupted by a mutated human beta globin intron 2. This mutation creates a new aberrant splicing site at position 705 that, when used, produced an mRNA that encodes a truncated non-functional luciferase (*Guidotti et al., 2017*). In the presence of the TAT-peptide nucleic acid (TAT-PNA) CPP described below, the aberrant splice site is masked allowing the production of a functional luciferase enzyme. Lentiviruses produced using this construct were employed to infect cells. The doses used resulted in >90% cells infected (based on GFP expression from the lentiviral vector). The infected cells (200,000 cells in 12-well plates containing 1 ml of RPMI, 10 % FBS) were treated or not with 5 µM TAT-PNA (GRKKRRQRRR-CCTCCTACCTCAGTTACA). TAT-PNA is made of TAT$_{48-57}$ and an oligonucleotide complimentary to a sequence containing the aberrant splice site. After 16 hr incubation, cells were washed twice in HKR buffer (119 mM NaCl, 2.5 mM KCl, 1 mM NaH$_2$PO$_4$, 2.5 mM CaCl$_2$, 1.3 mM MgCl$_2$, 20 mM HEPES, 11 mM dextrose, pH 7.4) and lysed in 40 µl HKR containing 0.1 % Triton X-100 for 15 min at room temperature. Luciferase activity was measured with a GLOMAXTM 96 Microplate Luminometer (Promega) using a Dual-Luciferase Reporter Assay (Promega) and normalized to the protein content. Results are displayed as the ratio between the protein-normalized luciferase signal obtained in TAT-PNA-treated cells and the signal obtained in control untreated cells.

## TAT-Cre recombinase production, purification, and recombination assay

Raji cells were infected with a lentivirus encoding a Cre-reporter gene construct (*D'Astolfo et al., 2015*). TAT-Cre recombinase was produced as described (*Wadia et al., 2004*). Briefly, *Escherichia coli* BL21 transformed with the pTAT-Cre plasmid (#917, Addgene plasmid #35619) were grown for 16 hr in LB containing 100 µg/ml kanamycin. Protein production was induced at OD600 of 0.6 with 500 µM IPTG (isopropyl β-D-1-thiogalactopyranoside) for 3 hr. Bacteria were collected by centrifugation at 5000 × *g* and kept at –20 °C. Purification was performed on Äkta prime (GE, Healthcare, Chicago, IL) equipped with a 1 ml HisTrap FF column equilibrated with binding buffer (20 mM sodium phosphate, 500 mM NaCl, 5 mM imidazole pH 7.4). The day of the purification, bacterial pellet was resuspended in lysis buffer (binding buffer with protease inhibitors; Roche, ref no. 4693132001; one tablet per 50 ml), 0.025 mg/ml DNase I (Roche, ref no. 04716728001), and 2 mg/ml lysozyme (Roche, ref no. 10 837 059 001) and sonicated six times for 30 s. After 20 min centrifugation at 5000 × *g*, the supernatant was filtered through Steriflip 0.45 µm and loaded on the column. Elution buffer (20 mM sodium phosphate, 500 mM NaCl, 500 mM imidazole pH 7.4) was used to detach His-tagged proteins from the column. Imidazole was removed from collected fractions by overnight dialysis using 10 K MWCO cassette (Thermo Scientific, ref no. 66807) in PBS. Raji cells encoding the Cre-reporter were treated for 48 hr with 20 µM TAT-Cre-recombinase. Fluorescence was imaged using a Nikon Eclipse TS100 microscope.

## Assessment of CPP binding to plasma membranes

Three-hundred thousand cells were incubated for 60 s in 1 ml RPMI supplemented with 10 % FBS and 10 mM HEPES in Eppendorf tubes at 37 °C in the presence of increasing concentrations of FITC-TAT-RasGAP$_{317-326}$. Half of the cells were then immediately placed on ice, pelleted at 4 °C, and resuspended in 1 ml of ice-cold PBS and then split into two tubes, one of which receiving a final concentration of 0.2 % (w:w) trypan blue to quench surface-associated FITC signals. The cells (still kept at 4 °C) were then analyzed by flow cytometry. The surface associated peptide signal was calculated by subtracting total fluorescence measured in PBS and fluorescence measured after trypan blue quenching. The other half of the cells after the 60 s peptide incubation was incubated at 37 °C for 1 hr at which time the cellular internalization of the labeled peptide was assessed by flow cytometry.

## Transient calcium phosphate transfection in HeLa cells

Calcium phosphate-based transfection of HeLa cells was performed as previously described (*Jordan et al., 1996*). Briefly, cells were plated overnight in DMEM (Invitrogen, ref no. 61965) medium supplemented with 10 % heat-inactivated FBS (Invitrogen, ref no. 10270–106), 2.5 µg of total plasmid DNA of interest was diluted in water, CaCl$_2$ was added and the mixture was incubated in presence of HEPES 2 × for 60 s before adding the total mixture drop by drop to the cells. Media was changed 10 hr after.

## Isothermal titration calorimetry

Isothermal titration calorimetry (ITC) was performed using MicroCal ITC200 (Malvern Panalytical) at 37 °C with 600 µM FITC-R9 in the cell (total volume 300 µl) and consecutive injections (2.5 µl/injection, except for the first injection of 0.4 µl) of 6 mM PI from the syringe (total volume 40 µl) with 2 min delay between injections and 800 rotations/min rotation speed. Differential power was set to 7, as we had no prior knowledge of the expected reaction thermodynamics. The results in *Figure 5—figure supplement 2A* are represented as a: thermogram (measurement of thermal power need to ensure that there is no temperature difference between reference and sample cells in the calorimeter as a function of time) and a binding isotherm (normalized heat per peak as a function of molar ratio).

## Colony formation assay

Three-hundred thousand wild-type HeLa cells were plated overnight in RPMI with 10 % FBS in six-well plates. Cells were then treated for 1 hr with the indicated concentrations of CPP, PI, and membrane potential modulating agents (gramicidin or valinomycin) in 1 ml RPMI. As control, cells were either left untreated or incubated with DMSO, the vehicle used to dilute gramicidin and valinomycin. Cells were then washed, trypsinized, and plated on 10 cm dishes at a density of 300 cells per condition. Colonies were counted at day 14 after 100 % ethanol fixation for 10 min and Giemsa staining. Washes were done with PBS.

## Genome-scale CRISPR/Cas9 knock-out screening

The human GeCKO v2 library (two plasmid system) (Addgene plasmid #1000000049) was amplified by electroporation using a Bio-Rad Gene Pulser II electroporation apparatus (Bio-Rad #165–2105) and the Lucigen Endura bacteria (Lucigen ref no. 60242). Cells were plated on LB agar plates containing 100 µg/ml ampicillin. After 14 hr at 32 °C, colonies were scrapped and plasmids recovered with the Plasmid Maxi kit (Qiagen, ref no. 12162). To produce lentivirus library, 12 T-225 flasks were seeded with $12 \times 10^6$ HEK293T cells per flask in 40 ml DMEM, 10 % FBS. The following day, 10 µg pMD2.G, 30 µg psPAX2, and 25 µg GeCKO plasmid library in 1.8 ml $H_2O$ were mixed with 0.2 ml 2.5 M $CaCl_2$ (final calcium concentration: 250 mM). This solution was mixed (v/v) with $2 \times$ HEPES buffer (280 mM NaCl, 10 mM KCl, 1.5 mM $Na_2HPO_4$, 12 mM D-glucose, 50 mM HEPES), incubated for 1 min at room temperature, added to the culture medium, and the cells placed back in a 37 °C, 5 % $CO_2$ incubator for 7 hr. The culture medium was then removed and replaced by DMEM supplemented with 10 % FBS containing 100 U/ml penicillin and 100 µg/ml streptomycin. Forty-eight hours later, the medium was collected and centrifuged 5 min at 2000× $g$ to pellet the cells. The remaining cell-free medium (12 × 40 ml) was then filtered through a 0.45 µm HV/PVDF (Millipore, ref no. SE1M003M00) and concentrated ~100 times by resuspending the viral pellet obtained by ultracentrifugation at 70000× $g$ for 2 hr at 4 °C in ~5 ml ice-cold PBS. The concentrated viruses were aliquoted in 500 µl samples and stored at –80 °C.

To express the Cas9 endonuclease, cells (e.g. Raji or SKW6.4) were infected with Cas9-expressing viruses that were produced in HEK293T cells transfected with the lentiCas9-Blast (#849, Addgene plasmid #52962), pMD2.G, and psPAX2 plasmids as described under 'Lentivirus production'. The infected cells were selected with 10 µg/ml blasticidin for a week. The multiplicity of infection (MOI) of the GeCKO virus library was determined as follows. Different volumes of the virus library were added to $3 \times 10^6$ Cas9-expressing cells plated in 12-well plates. Twenty-four hours later, the cells were split into two wells of 12-well plates. One well per pair was treated with 10 µg/ml puromycin for 3 days (the other cultured in normal medium). Cell viability was determined by trypan blue exclusion and MOI was calculated as the number of cells in the well treated with puromycin divided by the number of cells in the control well. The virus volume yielding to a MOI ~ 0.4 was chosen to perform large-scale infection of $12 \times 10^7$ cells that was carried out in 12-well plates with $3 \times 10^6$ cells per well. After 24 hr, the infected cells were collected and pooled in a T-225 flask and selected with 10 µg/ml puromycin for a week. Thirty millions of these were frozen (control untreated cells) and 60 million others were treated with 40 µM TAT-RasGAP$_{317-326}$ for 8 days (Raji) or for 17 days (SKW6.4) with a medium and peptide renewal every 2–3 days. Thirty million of the peptide-treated cells were then also frozen. Genomic DNA was extracted from the control and the peptide-treated frozen cells using the Blood & Cell Culture DNA Midi Kit according to manufacturer's instructions (Qiagen, ref no. 13343). A first PCR was performed to amplify the lentiCRISPR sgRNA region using the following primers:

> F1: 5'-AATGGACTATCATATGCTTACCGTAACTTGAAAGTATTTCG-3'
> R1: 5'-CTTTAGTTTGTATGTCTGTTGCTATTATGTCTACTATTCTTTCC-3

A second PCR (see *Supplementary file 4* for the primers used) was performed on 5 µl of the first PCR reaction to attach Illumina adaptors with barcodes (nucleotides highlighted in green) and to increase library complexity (using the sequences highlighted in red) to prevent signal saturation when the sequencing is performed. The blue sequences are complementary to the extremities of the first PCR fragments.

Both PCRs were performed in 100 µl with the 2 µl of the Herculase II Fusion DNA Polymerase from Agilent (ref no. 600675) according the manufacturer's instructions. Amplicons were gel extracted, quantitated, mixed, and sequenced with an MiSeq (Illumina). Raw FASTQ files were demultiplexed and processed to contain only unique sgRNA sequences. The number of reads of each sgRNA was normalized as described (*Tribello et al., 2014*). The MAGeCK algorithm (*Tsoutsou et al., 2017*) was used to rank screening hits by the consistent enrichment among multiple sgRNAs targeting the same gene.

## CRISPR/Cas9-based genome editing

Single guide RNAs targeting an early exon of the gene of interest were chosen in the sgRNA library (*Wallbrecher et al., 2017*) and are listed in *Supplementary file 5*. LentiCRISPR plasmids specific

for a gene were created according to the provided instructions (*Vasconcelos et al., 2013*). Briefly, oligos were designed as follows: Forward 5′-CACCGnnnnnnnnnnnnnnnnnnnn-3′; Reverse-3′-CnnnnnnnnnnnnnnnnnnnnnCAA-5′, where nnnnnnnnnnnnnnnnnnnn in the forward oligo corresponds to the 20 bp sgRNA. Oligos were synthetized, then phosphorylated, and annealed to form oligo complexes. LentiCRISPR vector was BsmBI digested and dephosphorylated. Linearized vector was purified and gel extracted and ligated to oligo complexes. The lentiCRISPR vector containing the sgRNA was then used for virus production. Recombinant lentiviruses were produced as described (*Mitchell et al., 2000*) with the following modification: pMD.G and pCMVDR8.91 were replaced by pMD2.G and psPAX2, respectively. Cells were infected and selected with the appropriate dose of puromycin (2 µg/ml for HeLa cells). Clone isolation was performed by limiting dilution in 96-well plate.

## TA cloning
TA cloning is a subcloning technique that allows integration of a PCR-amplified product of choice into a PCR2.1 vector based on complementarity of deoxyadenosine added onto the PCR fragment by Taq polymerase. This approach is useful to distinguish between several alleles and to determine whether the cells are heterozygous or homozygous at a given locus. TA cloning kit (Life Technologies, ref no. K202020) was used according to manufacturer's instructions to sequence DNA fragment containing the region targeted by a given sgRNA. Briefly, DNA was isolated and the fragment of interest was PCR-amplified using primers listed in *Supplementary file 6*, then ligated into PCR2.1 vector. *E. coli* competent cells were then transformed and at least 15 colonies were selected per condition for DNA isolation and sequencing.

## Plasmid constructs
The hKCNN4-V5.lti (#953) lentiviral plasmid encoding a V5-labeled version of the KCNN4 potassium channel was from DNASU (ref no. HsCD00441560). The hKCNK5-FLAG.dn3 (#979) plasmid encoding the human KCNK5 potassium channel (NCBI reference sequence NM_003740.3), Flag-tagged at the C-terminus, was purchased from GenScript (ref no. OHu13506). The Myc-mKCNJ2-T2A-IRES-tdTo-mato.lti (#978) lentiviral vector encoding the mouse Kir2.1 (KCNJ2) potassium channel and tdTo-mato (separated by an IRES) was generated by subcloning myc-mKCNJ2-T2A-Tomato.pCAG plasmid (#974, Addgene plasmid #60598) into the LeGo-iT2 lentiviral backbone (#809), a gift from Boris Fehse (Addgene plasmid #27343), through ligation of both plasmids after digestion with BamHI (NEB, reg. no. R313614). The pMD2.G plasmid (#554, Addgene plasmid #12259) encodes the envelope of lenti-virus. The psPAX2 plasmid (#842, Addgene plasmid #12260) encodes the packaging system. Both pMD2.G and psPAX2 plasmids were used for lentiviral production. The Flag-hKCNQ5(G278S)-IRES-NeoR plasmid (#938) codes for the N-terminal Flag-tagged G278S human KCNQ5 inactive mutant and a neomycin resistant gene separated by an IRES sequence. It was generated by subcloning a BamHI/XmaI digested PCR fragment obtained by amplification of pShuttle-Flag-hKCNQ5(G278S)-IRES-hrGFP2 (#937, kind gift from Dr Kenneth L Byron) using forward primer #1397 (CAT CGG GAT CCG CTA TAC CGG CCA CCA TGG ATT ACA AGG A) and reverse primer #1398 (CAT CGC CCG GGG CTA TAC CGT ACC GTC GAC TGC AGA ATT C) into the lentiviral vector TRIP-PGK-IRES-Neo (#350) opened with the same enzyme. The Flag-hKCNQ5(SM,G278S)-IRES-Neo (#939) plasmid is identical to Flag-hKCNQ5(G278S)-IRES-NeoR except that the sequence targeted by the sgKCNQ5.1 sgRNA (*Supplementary file 5*) was mutated with the aim to decrease Cas9-mediated degradation. Silent mutations (SM), at the protein level, were introduced using the QuikChange II XL Site-Directed Mutagenesis Kit (ref no. 200522) according to manufacturer's instructions using forward primer #1460 (AAA TAA GAA CCA AAA ATC CTA TGT ACC ATG CCG TTA TCA GCT CCT TGC TGT GAG CAT AAA CCA CTG AAC CCA G) and reverse primer #1461 (CTG GGT TCA GTG GTT TAT GCT CAC AGC AAG GAG CTG ATA ACG GCA TGG TAC ATA GGA TTT TTG GTT CTT ATT T).

The Flag-hKCNQ5(SM)-IRES-NeoR (#940) lentiviral construct codes for a Flag-tagged wild-type version of human KCNQ5. It was made by reverting the G278S mutation in Flag-hKCNQ5(SM,G278S)-IRES-Neo (#939) using the QuikChange II XL Site-Directed Mutagenesis Kit with the #1462 forward primer (TTT TGT CTC CAT AGC CAA TAG TTG TCA ATG TAA TTG TGC CCC) and the #1463 reverse primer (GGG GCA CAA TTA CAT TGA CAA CTA TTG GCT ATG GAG ACA AAA). The pLUC705 (*Kang et al., 1998*) (#876, gift from Dr Bing Yang) plasmid encodes a luciferase gene interrupted by a mutated human beta globin intron 2. This mutation creates a new aberrant splicing site at position

705 that when used produced an mRNA that encodes a truncated non-functional luciferase (*Kang et al., 1998*). To introduce this construct into a lentiviral vector, the pLUC705 plasmid was digested with HindIII/XhoI, blunted with T4 DNA polymerase, and ligated into StuI-digested and dephosphorylated LeGO-iG2 (#807, Addgene plasmid #27341), resulting in plasmid pLUC705.LeGO-iG2 (#875). The pTAT-Cre (#917, Addgene plasmid #35619) bacterial plasmid encodes a histidine-tagged TAT-Cre recombinase. The Cre reporter lentiviral vector (#918, Addgene plasmid #62732) encodes a LOXP-RFP-STOP-LOXP-GFP gene construct. Cells expressing this plasmid appear red but once recombination has occurred when TAT-Cre is translocated into cells, the RFP-STOP fragment will be excised, GFP but not RFP will now be produced, and cells will appear green.

## Peptides

TAT-RasGAP$_{317-326}$ is a retro-inverso peptide (i.e. synthesized with D-amino-acids in the opposite direction compared to the natural sequence) labeled or not with FITC or TMR. The TAT moiety corresponds to amino acids 48–57 of the HIV TAT protein (RRRQRRKKRG) and the RasGAP$_{317-326}$ moiety corresponds to amino acids from 317 to 326 of the human RasGAP protein (DTRLNTVWMW). These two moieties are separated by two glycine linker residues in the TAT-Ras-GAP$_{317-326}$ peptide. FITC-bound peptides without cargo: TAT, Penetratin (RQIKWFQNRRMKWKK), R9 (RRRRRRRRR), and K9 (KKKKK-KKKK) were synthesized in D-amino acid conformation. All peptides were synthesized in retro-inverso conformation (over the years different suppliers were used with routine checks for activity of TAT-RasGAP$_{317-326}$ derived peptides, Biochemistry Department of University of Lausanne, SBS Genetech, China and Creative Peptides, USA) and resuspended to 1 mM in water.

## Statistical analysis

Statistical analysis was performed on non-normalized data, using GraphPad Prism 7. ANOVA multiple comparison analysis to wild-type condition was done using Dunnett's correction (*Figure 3A–C* and *Figure 3—figure supplement 1B, C* and *Figure 3C*, top panel, and PI internalization in *Figure 5C*, as well as TAT-PNA internalization in *Figure 2—figure supplement 4A*). ANOVA multiple comparison analysis between several conditions was done using Tuckey's correction (*Figure 3—figure supplement 2E*). Comparison between two conditions was done using two-tailed paired t-test for the CPP internalization experiments described in *Figure 3C* (bottom panel), *Figure 6A–B*, *Figure 1—figure supplement 1B*, and *Figure 3—figure supplement 4*. All measurements were from biological replicates. Unless otherwise stated, the horizontal bars in the graph represent the median, the height of columns correspond to averages, and the dots in the figures correspond to values derived from independent experiments.

## Acknowledgements

This work was supported by grants from the Swiss National Science Foundation awarded to CW (no. CRSII3_154420, IZCSZ0-174639), to NC (no. 158116), to FA (no. 320030_170062) and to AL (no. 310030-184759). We are thankful to Prof. Denise Nardelli Haefliger and her group at the Lausanne University Hospital, especially to Sonia Domingos Pereira and Laurent Derre for helpful discussions. We are thankful to the Cellular Imaging Facility, Mouse Pathology Facility and Genomic Technologies Facility at the University of Lausanne for the resources provided and their technical help. We would like to thank Giacomo Nanni for his technical help in running the molecular simulations. We are also thankful to the Swiss National Supercomputing Centre (CSCS). We are grateful to Jacques Morisod for helping us perforing the ITC experiments at the Laboratoires des Polymères (EPFL, Lausanne, Switzerland).

# Additional information

## Funding

| Funder | Grant reference number | Author |
|---|---|---|
| Swiss National Science Foundation | CRSII3_154420 IZCSZ0-174639 | Christian Widmann |
| Swiss National Science Foundation | 158116 | Nadja Chevalier |
| Swiss National Science Foundation | 320030_170062 | Francesca Amati |
| Swiss National Science Foundation | 310030-184759 | Anita Lüthi |

The funders had no role in study design, data collection and interpretation, or the decision to submit the work for publication.

## Author contributions

Evgeniya Trofimenko, Conceptualization, Formal analysis, Investigation, Methodology, Validation, Visualization, Writing - original draft, Writing - review and editing; Gianvito Grasso, Conceptualization, Formal analysis, Investigation, Methodology, Software, Validation, Visualization; Mathieu Heulot, Nadja Chevalier, Conceptualization, Investigation, Validation; Marco A Deriu, Software; Gilles Dubuis, Linh Chi Dam, Investigation, Validation; Yoan Arribat, Sebastien Michel, Florine Ory, Investigation; Marc Serulla, Formal analysis, Investigation; Gil Vantomme, Formal analysis, Supervision; Julien Puyal, Francesca Amati, Anita Lüthi, Andrea Danani, Resources, Supervision; Christian Widmann, Conceptualization, Formal analysis, Funding acquisition, Project administration, Resources, Supervision, Visualization, Writing - original draft, Writing - review and editing

## Author ORCIDs

Evgeniya Trofimenko (iD) http://orcid.org/0000-0003-4910-5324
Gianvito Grasso (iD) http://orcid.org/0000-0002-7761-222X
Mathieu Heulot (iD) http://orcid.org/0000-0001-9641-941X
Marco A Deriu (iD) http://orcid.org/0000-0003-1918-1772
Yoan Arribat (iD) http://orcid.org/0000-0003-0952-5279
Marc Serulla (iD) http://orcid.org/0000-0002-4166-1556
Gil Vantomme (iD) http://orcid.org/0000-0002-7441-0737
Linh Chi Dam (iD) http://orcid.org/0000-0002-6874-2049
Julien Puyal (iD) http://orcid.org/0000-0002-8140-7026
Francesca Amati (iD) http://orcid.org/0000-0002-1731-0262
Anita Lüthi (iD) http://orcid.org/0000-0002-4954-4143
Andrea Danani (iD) http://orcid.org/0000-0001-6578-5089
Christian Widmann (iD) http://orcid.org/0000-0002-6881-0363

## Ethics

All experiments were performed according to the principles of laboratory animal care and Swiss legislation under ethical approval (Swiss Animal Protection Ordinance; permit number VD3374.a).

## Decision letter and Author response

Decision letter https://doi.org/10.7554/eLife.69832.sa1
Author response https://doi.org/10.7554/eLife.69832.sa2

# Additional files

## Supplementary files

• Supplementary file 1. Components of the culture medium lacking potassium chloride and sodium chloride. This table lists the components found in the Biowest RPMI-like media that lacks potassium chloride and sodium chloride.

• Supplementary file 2. Membrane lipid composition considered in the present study. This table lists

the proportions of various lipids found in the inner and outer layers of the plasma membrane that we have used for our simulations.

• Supplementary file 3. Nascent water pore free energy estimations in various studies. This table reports the free energy that needs to be overcome for the formation of water pores that was calculated in the indicated studies.

• Supplementary file 4. Primer sequences used in the second PCR performed to bar code the single guide RNAs (sgRNAs) of the selected populations. The green nucleotides correspond to the bar codes. The red nucleotides are added to increase library complexity to prevent signal saturation when the sequencing is performed. The blue sequences are complementary to the extremities of the first PCR fragments.

• Supplementary file 5. Single guide RNAs (sgRNAs) used to disrupt the indicated genes. This table provides the sequences of the sgRNAs used to target the first exon of the indicated genes.

• Supplementary file 6. Forward and reverse primers used in the TA cloning procedure.

• Transparent reporting form

### Data availability

DNA sequencing data from the CRISPR/Cas9-based screens are available through the following link: https://www.ncbi.nlm.nih.gov/sra/SRP161445.

The following dataset was generated:

| Author(s) | Year | Dataset title | Dataset URL | Database and Identifier |
|---|---|---|---|---|
| Trofimenko E, Grasso G, Heulot M, Chevalier N, Deriu MA, Dubuis G, Arribat Y, Serulla M, Michel S, Vantomme G, Ory F, Dam LC, Puyal J, Amati F, Lüthi A, Danani A, Widmann C | 2018 | CRISPR/Cas9 screen to identify genes involved in uptake of cell-penetrating peptides | https://www.ncbi.nlm.nih.gov/sra/SRP161445 | NCBI Sequence Read Archive, SRP161445 |

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
