## [Editor Report]

Although the role of membrane potential in cell-penetrating peptide (CPP) translocation has been consistently described in artificial systems, this multi scale study combining cell biology, genetics and in silico approaches further extends this topic to a live cell context where it shows that internalization stops when the membrane polarization is decreased by the removal of potassium channels. It proposes an original mechanism of CPP translocation based on transient water pore formation, which should be of interest for biophysicists, cell biologists and for applications such as drug deliver.

---

## [Decision Letter]

**Decision letter after peer review:**

Thank you for submitting your article "Genetic, cellular and structural characterization of the membrane 1 potential-dependent cell-penetrating peptide translocation pore" for consideration by *eLife*. Your article has been reviewed by 2 peer reviewers, and the evaluation has been overseen by a Reviewing Editor and Olga Boudker as the Senior Editor. The following individual involved in review of your submission has agreed to reveal their identity: Alain Joliot (Reviewer #2).

Essential revisions:

We believe that your study is original and brings new data to the field. The additional data that we ask should strengthen your paper and should be feasible in about 2 months.

1) Data analysis:

Quantification of CPP uptake that takes into account the intracellular localization of the CPP (cytosolic versus vesicular) is a critical issue in cellular models. In this study, quantification relies either on the toxicity induced by the cytosolic accumulation of the Tat-RasGAP peptide or on the direct visualization of fluorescently-labelled CPPs. In the latter case, the choice made by the authors to define 3 categories for quantification has to be justified. More precisely, merging the low and strong cytosolic signal categories into a single one is questionable as these two categories might correspond to distinct functional fates due to the highly variable cytosolic staining (much more than between vesicular and low cytosolic). Indeed, the water pore mechanism invoked by the authors might correspond to an "all or none" mechanism for the cytosolic delivery that thus, to low and high cytosolic contents, respectively. It would be then relevant to determine how each category (low and high) is affected by the membrane potential.

In order to link functional (toxicity) and live imaging analysis assays, one possibility would be to analyze the remaining living cells following long incubation (16-24h) with FITC-TAT RasGAP to estimate which staining category is actually killed (ideally with or without hyperpolarization). The non-toxic mutant TatRASGAP (W317A) could be used as control.

Optional: To check whether the very large heterogeneity in the translocation efficiency within a same cell culture could be related to heterogeneities in membrane potential among cells, the membrane potential of each cell could be measured (using DiBAC4(3) labelling or any genetically encoded membrane potential sensor) together with CPP translocation (labelled TMR if using DiBAC).

2) Ectopic expression of the potassium channel in "heterologous" context.

Although distinct types of potassium channels have been characterized in the screen, the authors only consider their action on the membrane potential, supported by the effects of specific drugs. However, translocation efficiency does not strictly correlate with membrane potential (e. g. KCNJ2 expression in WT and KCNN4 KO Hela figure 3B). It would be interesting to evaluate if KCNQ5 expression would rescue or even increase internalization (additive effect) in SW6.4 and HeLa cells (WT and KCNN4 KO) and vice versa (KCNN4 expression in Raji). This would also avoid any potential interference of the CRISPR system on ectopic expression. Indeed, the kinetic of CPP uptake significantly differs between cell lines (Figure 1B, almost 100% negative cells for Raji at 20 minutes and only 20% for the 2 others), suggesting partially distinct mechanisms.

On the simulations:

3)The limitations of the simulations should be discussed. The main limitation is the structure of peptides because the Martini forcefield (the version should be mentioned) is not able to properly capture peptide folding, which can be important in membrane adsorption/translocation. Thus, the in silico results should be viewed only as an indication (not proof), the folding issue should be mentioned in the manuscript and the authors should more clearly specify what the peptide structures are.

4) Current simulations seem to mainly capture the effect of simple electrostatics; i.e. more charges, more penetration. To test this, the authors could do simulations with K9 and compare it to R9, which is known experimentally to translocate better than K9. Such comparison would show what simulations actually detect.

Optional:

– Further check experimentally the physical mechanism related to the membrane potential: compare the uptake effectiveness for different lengths of poly-arginines. If the mechanism is correct, the effectiveness will increase with increasing length.

[Editors' note: further revisions were suggested prior to acceptance, as described below.]

Thank you for submitting your article "Genetic, cellular and structural characterization of the membrane potential-dependent cell-penetrating peptide translocation pore" for consideration by *eLife*. Your article has been reviewed by 2 peer reviewers, and the evaluation has been overseen by a Reviewing Editor and Olga Boudker as the Senior Editor. The following individuals involved in review of your submission have agreed to reveal their identity: Alain Joliot (Reviewer #2).

Essential revisions:

*Reviewer #1 (Recommendations for the authors):*

The authors responded to all the raised issues and modified the manuscript accordingly. The manuscript represents an important contribution to the field of CPP translocation into cells. Based on the experiments and simulations CPP, they propose a mechanism that CPPs enter the cell thanks to the electrostatic membrane potential. The data are well presented, however, it seems to me that the authors overinterpret the data because the presented mechanism and its discussion have few flaws:

On page 21: The authors state: "However, it has been determined that, once embedded into membranes, lysine residues tend to lose protons (Armstrong et al., 2016; Li et al., 2013). This will thus dissipate the strong membrane potential required for the formation of water pores and leave the lysine-containing CPPs stuck within the phospholipids of the membrane."

Lysine can indeed get neutral in the membrane, and the deprotonated state is the most favorable state in the middle of the membrane hydrophobic core (see MacCallum et al., 2008 for free energy profiles of lysine insertion). However, it is energetically unfavorable for lysine to be inside the membrane even when deprotonated. The most favorable state for lysin is to be protonated in solution (around neutral pH), i.e., lysine residue is hydrophilic, not hydrophobic. Therefore, lysine residues preferentially leave the membrane and not get stuck inside the membrane, as the authors state. Moreover, the proton will dissociate during the lysine insertion into the membrane, and thus it most likely stays outside of the cell. Note that when peptides are in the pore, they are not in the hydrophobic core of the membrane.

Page 22: "…water pores created by megapolarization have a diameter of about 2(-5) nm. Molecules larger than 2 nm are therefore less efficiently transported through these water pores. Polyarginine peptides of 20 amino acids or more have a predicted spherical diameter greater than 2 nm and would travel less efficiency through the water pores than shorter peptides. The efficiency of direct translocation is therefore likely modulated by the number of positive charges of the peptide and the size of the peptides (not mentioning the role of the secondary structures adopted by the CPPs)."

I agree with the general statement that the uptake of large molecules could be hindered or even prevented by the limited size of the pore size. However, the PEPFOLD-3 server, which the authors used for the prediction of peptide structures fixed in all simulations, predicts helical structure for polyarginine peptides with lengths 9 and even 20. Such helix has a diameter smaller than 2 nm along the helical axes and thus can easily get through the pore. Therefore, the presented data/discussion is not consistent with simulations.

Finally, the authors correctly state that arginine residues could be replaced by tryptophan residues in CPPs leading to similar translocation in cells. This possibility clearly demonstrates that there is more into the CPP translocation mechanism than the electrostatic membrane potential. The used explanation "It appears that loss of positive charges that contribute to water pore formation can be compensated by acquisition of strengthened lipid interactions when arginine residues are replaced with tryptophan residues." is strange because the use of more residues with less hydrophobic character would then have the same effect (e.g., two phenylalanine residues instead of arginine or tryptophan). In addition, the membrane potential does not explain the experiments on GUVs. Therefore, despite a nice demonstration of the importance of the membrane potential for the CPP translocation into cells, the mechanism seems to be more complex.

*Reviewer #2 (Recommendations for the authors):*

I really appreciate the efforts of the authors to provide new experimental data in response to the comments, which reinforce the quality of the manuscript. I fully agree that the time component should be critical at the level the whole cell population, which is not incompatible with an all or none mechanism at the single cell level, i.e.once reaching the favorable membrane potential. Surprisingly, fluorescent median intensity rather than the percentage of cells with high intensity signal seems to increase over time (Figure 1 sup4, error in axis title panel D) at least in HeLa cells. Inherent to most original studies, this work raises some interesting questions which remain unanswered or unclear but this would be part of another study. In particular, the use of genetic sensor instead of diBac to measure membrane potential might greatly help to dissect the translocation process on a single-cell basis.

In conclusion I would recommend the publication of this manuscript in its revised version

---

## [Author Response]

Essential revisions:We believe that your study is original and brings new data to the field. The additional data that we ask should strengthen your paper and should be feasible in about 2 months.1) Data analysis:Quantification of CPP uptake that takes into account the intracellular localization of the CPP (cytosolic versus vesicular) is a critical issue in cellular models. In this study, quantification relies either on the toxicity induced by the cytosolic accumulation of the Tat-RasGAP peptide or on the direct visualization of fluorescently-labelled CPPs. In the latter case, the choice made by the authors to define 3 categories for quantification has to be justified. More precisely, merging the low and strong cytosolic signal categories into a single one is questionable as these two categories might correspond to distinct functional fates due to the highly variable cytosolic staining (much more than between vesicular and low cytosolic). Indeed, the water pore mechanism invoked by the authors might correspond to an "all or none" mechanism for the cytosolic delivery that thus, to low and high cytosolic contents, respectively. It would be then relevant to determine how each category (low and high) is affected by the membrane potential.

We thank the reviewer for this interesting comment. The reviewer suggests that water pore formation mechanism is an “all or none” process, which could be the case. However, based on our results in Figure 1, another plausible interpretation to account for CPP uptake heterogeneity within the cell population is the difference in kinetics of water pore formation, indicating a need for a certain membrane potential threshold favorable to water pore formation. We have seen that over time there are more and more cells that acquire the CPP within the cytosol, which supports the time component of the process.

As the reviewer suggested, we have now quantitated the percentage of cells with different CPP fluorescence intensity within the cytosol in non-treated, depolarized (gramicidin treatment) or hyperpolarized (valinomycin treatment) conditions (see Figure 3 – figure supplement 2D ). As seen previously (Figure 3C-D), Penetratin uptake is slightly less affected by membrane potential modulation than the uptake of TAT and R9, though the trend remains the same. For all three tested CPPs, in the control condition, there is a mixed population of cells where the CPP cytosolic signal is either absent, low or high. The fact that there is a mix of the CPP cytosolic uptake efficiency in the non-treated population could be due to differences in the basal membrane potential within the cell population. Upon depolarization, cells with high cytosolic signal were no longer observed. Most cells were actually unable to acquire the CPPs in their cytosol. However, when cells were hyperpolarized by valinomycin, the percentage of cells without cytosolic signal became negligible and the number of cells with high cytosolic CPP signal increased in comparison to the non-treated condition. These results confirm our previous results that cytosolic CPP access is membrane-potential dependent, which is reflected in the increased intensity of CPP cytosolic signal in hyperpolarized conditions. These results are now part of Figure 3 – figure supplement 2D.

In order to link functional (toxicity) and live imaging analysis assays, one possibility would be to analyze the remaining living cells following long incubation (16-24h) with FITC-TAT RasGAP to estimate which staining category is actually killed (ideally with or without hyperpolarization). The non-toxic mutant TatRASGAP (W317A) could be used as control.

We now provide an additional experiment to address this question in [Figure 1 —figure supplement 4] by combining flow cytometry and confocal microscopy data. Despite intensity differences, the appearance of the populations with high and low CPP fluorescence occurred with similar kinetics (panels A-B). There was an almost perfect correlation between:

i) the percentage of cells with low intensity peptide fluorescence detected by flow cytometry and the percentage of cells with vesicular staining and low CPP cytosolic signal visualized with confocal microscopy (panel C),

ii) the percentage of cells with high intensity peptide fluorescence detected by flow cytometry and the percentage of cells with high CPP cytosolic staining visualized with confocal microscope (panel D).

Cell death induced by TAT-RasGAP_317-326_ was mostly recorded in cells from the high-intensity fluorescent population (panel E), which, based on our results, corresponds to cells that had acquired the CPP through direct translocation across the cell membrane. Hence, cytosolic accumulation of the peptide in relatively high amounts is what allows TAT-RasGAP_317-326_ to kill HeLa cells. For the mechanism underlying the killing activity of TAT-RasGAP_317-326_, and why the TAT-RasGAP_317-326_ mutant lacks this activity, please refer to the following reference: Serulla et al. PNAS 2020 117:31871.

Optional: To check whether the very large heterogeneity in the translocation efficiency within a same cell culture could be related to heterogeneities in membrane potential among cells, the membrane potential of each cell could be measured (using DiBAC4(3) labelling or any genetically encoded membrane potential sensor) together with CPP translocation (labelled TMR if using DiBAC).

This is indeed an interesting point. We, as well as others^1-7^, have regularly observed different abilities of cells within a culture to take up CPPs. Some take up the CPP by endocytosis, others by direct translocation, many using both eventually. There can also be cells in a population that do not take up CPPs at all. The observed variability is currently not understood and could be modulated by the metabolic state of the cell, by the extent of cell polarization, as well as in which stage of the cell cycle cells are (for the latter point, it is known indeed that cells close to mitosis become refractory to endocytosis). In our DiBac3(4) experiments performed with control cells, we have seen that the probe fluorescence follows a relatively broad Gaussian distribution (i.e. spanning about 1 ½ logs of fluorescence intensity) indicating that cells within a population can have very different Vm (see Author response image 1). Similar results were recently obtained using single cell membrane potential detection that showed that within a cell population the Vm vary between -100 mV to 0 mV ^8^. We have attempted to co-incubate the cells with TMR-CPP and DiBac3(4). However, these two molecules seem to interact with each other, preventing adequate interpretation of the results. Nevertheless, we agree with the reviewer that the heterogeneity in the CPP uptake could reflect the membrane potential differences within the cell population.

**Author response image 1. sa2fig1:** 

2) Ectopic expression of the potassium channel in "heterologous" context.Although distinct types of potassium channels have been characterized in the screen, the authors only consider their action on the membrane potential, supported by the effects of specific drugs. However, translocation efficiency does not strictly correlate with membrane potential (e. g. KCNJ2 expression in WT and KCNN4 KO Hela figure 3B). It would be interesting to evaluate if KCNQ5 expression would rescue or even increase internalization (additive effect) in SKW6.4 and HeLa cells (WT and KCNN4 KO) and vice versa (KCNN4 expression in Raji). This would also avoid any potential interference of the CRISPR system on ectopic expression. Indeed, the kinetic of CPP uptake significantly differs between cell lines (Figure 1B, almost 100% negative cells for Raji at 20 minutes and only 20% for the 2 others), suggesting partially distinct mechanisms.

We thank the reviewer for this interesting suggestion of potassium channel expression in “heterologous” context. Through these experiments (see Figure 3 —figure supplement 1) we have indeed been able to show that the KCNQ5 and KCNN4 CRISPR/Cas9-identified potassium channels modulate the cell membrane potential not only when present endogenously, but also when ectopically expressed. Ectopic expression of the identified potassium channels led to cell hyperpolarization (albeit to a lesser extent than what was observed in Figure 3B), which as expected resulted in increased CPP internalization. These results are now part of Figure 3 —figure supplement 1C.

The reviewer is correct that CPP internalization in Raji cells is somewhat delayed in comparison to HeLa and SKW6.4 cells. It is possible that distinct mechanisms are used by different cell lines to acquire CPPs in their cytosol. However, since direct translocation, in all the tested cell lines, is inhibited by depolarization and increased by hyperpolarization, we rather suggest that direct translocation follows the mechanism described in our manuscript but with different kinetics and efficiencies between cell lines possibly due to differences in cell membrane composition affecting initial peptide surface binding and consequently modulating further water pore formation and CPP entry.

On the simulations:3) The limitations of the simulations should be discussed. The main limitation is the structure of peptides because the Martini forcefield (the version should be mentioned) is not able to properly capture peptide folding, which can be important in membrane adsorption/translocation. Thus, the in silico results should be viewed only as an indication (not proof), the folding issue should be mentioned in the manuscript and the authors should more clearly specify what the peptide structures are.

Martini forcefield, as any other model, has indeed a number of known limitations^9,10^ such as the chemical and spatial resolution, which are both limited compared to atomistic models. There is also a shifted balance between entropy and enthalpy due to the reduced number of degrees of freedom. Moreover, as correctly mentioned by the Reviewer, the secondary structure is an input parameter of the model, which implies that secondary structure elements remain fixed during the simulation^11^, starting from the molecular model obtained through the PEPFOLD-3 server^12-14^. Tertiary structural changes, however, are allowed and in principle realistically described with Martini forcefield. The coarse-grained approach has demonstrated to provide reliable results in the context of protein translocation mechanism^15^, estimation of tilt angle of membrane spanning helices^11^, or helix-helix packing motifs^16,17^, H-NMR quadrupolar splitting of WALP peptides (reproduced to a better extent than atomistic simulations)^18^, structure prediction of the glycophorin A dimer compared to NMR data^16^, water transmembrane (TM) permeation rate^11,19^, propensity for interfacial versus TM peptide orientation^11^, lateral diffusion of peptides^20^ and proteins^21^, and dimerization free energies of transmembrane helices^22^. Moreover, the ability of Martini Coarse-grained forcefield to model realistic, heterogeneous, membranes was widely demonstrated in literature. Extensive comparison of the performance of the Martini model with respect to a variety of experimental properties has revealed that the model performs generally very well in term of membrane thickness, area per lipid (typically reproduced to within 0.1–0.2 nm^2^)^19,23^, ternary phase behavior of lipid mixtures^24^, liquid densities^19^, accessible lipid conformations^25^, membrane bending modulus^19^, rupture tension^23^, diffusion rates of lipids^19,23^, time scales for lipid aggregation^19^, bilayer phase transition temperatures^26,27^, lipid desorption free energy^23^, membrane domain formation^24,28^ data. The scientific literature in the field of coarse-grained molecular simulations of cell membranes is summarized in a recent review^10^. In the context of electroporation, the implementation of a polarizable water model for the coarse-grained MARTINI force field^19,29^ allowed to induce pores in a membrane in a similar manner as it was done in atomistic simulations^19,30^. However, treatment of long-range electrostatic interactions with the PME method is a prerequisite to observe pores and therefore an increased computational effort is needed. For this reason, we applied the PME method to treat the electrostatic interactions (Please see the Material and Method section of the manuscript).

We’ve carefully discussed the limitations of the coarse-grained approach applied in the manuscript, as required by the Reviewer in the Discussion section of the manuscript. Moreover, we’ve also specified the version of the forcefield used. We would also specify that, as reported in Material and Methods section of the manuscript, the CPP molecular structure has been generated through the PEPFOLD-3 server^12^, as done in recent manuscripts in the field^13,31^.

4) Current simulations seem to mainly capture the effect of simple electrostatics; i.e. more charges, more penetration. To test this, the authors could do simulations with K9 and compare it to R9, which is known experimentally to translocate better than K9. Such comparison would show what simulations actually detect.

The reviewer is correct and we mention in the discussion that, according to our model, K9 should induce megapolarization and formation of water pores. This should then allow efficient K9 translocation into cells, contradicting the cellular observations made by us and others that K9 internalization if less efficient than that of R9 (Figure 3 —figure supplement 2C). It has been determined that besides the differences in the formation of hydrogen bonds between the two CPPs, lysine and arginine differ in their protonation state. Once embedded into membranes, lysine residues tend to lose protons^32,33^. This will thus dissipate the strong membrane potential required for the formation of water pores and leave the lysine-containing CPPs stuck within the phospholipids of the membrane. In contrast, arginine residues are not deprotonated in membranes and water pores can therefore be maintained allowing the arginine-rich CPPs to be taken up by cells. This phenomenon however cannot be modeled by coarse-grained *in silico* simulations because the protonation state is fixed at the beginning of the simulation runs and is not allowed to evolve. Therefore, the uptake kinetics of lysine-rich peptides, such as MAP and K9, would appear artefactually similar as the uptake kinetics of arginine-rich peptides such as R9. This was discussed in the previous version of the manuscript.

Optional:– Further check experimentally the physical mechanism related to the membrane potential: compare the uptake effectiveness for different lengths of poly-arginines. If the mechanism is correct, the effectiveness will increase with increasing length.

Several previously published studies such as^34,35^ have addressed this question by comparing the internalization efficiency of polyarginine peptides of various lengths. According to these studies the optimal length of consecutive arginine residues appears to be between 9-16 amino acids, resulting in optimal CPP cytosolic acquisition. Shorter and longer peptides have decreased internalization efficiencies. The role of the Vm presented in our model is consistent with the reduced uptake of short polyarginine peptides but the Vm parameter of our model cannot explain why longer polyarginine peptides are less efficiently taken up by cells. Our work however also indicates that the water pores created by megapolarization have a diameter of about 2(-5) nm. Molecules larger than 2 nm are therefore less efficiently transported through these water pores. Polyarginine peptides of 20 amino acids or more have a predicted spherical diameter greater than 2 nm and would travel less efficiency through the water pores than shorter peptides. The efficiency of direct translocation is therefore likely modulated by the number of positive charges of the peptide and the size of the peptides (not mentioning the role of the secondary structures adopted by the CPPs). This is now addressed in the discussion of the updated version of the manuscript.

[Editors' note: further revisions were suggested prior to acceptance, as described below.]

Essential revisions:Reviewer #1 (Recommendations for the authors):The authors responded to all the raised issues and modified the manuscript accordingly. The manuscript represents an important contribution to the field of CPP translocation into cells. Based on the experiments and simulations CPP, they propose a mechanism that CPPs enter the cell thanks to the electrostatic membrane potential. The data are well presented, however, it seems to me that the authors overinterpret the data because the presented mechanism and its discussion have few flaws:On page 21: The authors state: "However, it has been determined that, once embedded into membranes, lysine residues tend to lose protons (Armstrong et al., 2016; Li et al., 2013). This will thus dissipate the strong membrane potential required for the formation of water pores and leave the lysine-containing CPPs stuck within the phospholipids of the membrane."Lysine can indeed get neutral in the membrane, and the deprotonated state is the most favorable state in the middle of the membrane hydrophobic core (see MacCallum et al., 2008 for free energy profiles of lysine insertion). However, it is energetically unfavorable for lysine to be inside the membrane even when deprotonated. The most favorable state for lysin is to be protonated in solution (around neutral pH), i.e., lysine residue is hydrophilic, not hydrophobic. Therefore, lysine residues preferentially leave the membrane and not get stuck inside the membrane, as the authors state. Moreover, the proton will dissociate during the lysine insertion into the membrane, and thus it most likely stays outside of the cell. Note that when peptides are in the pore, they are not in the hydrophobic core of the membrane.

We thank the reviewer for raising the point that lysine-containing peptides are unlikely to get “stuck” in the plasma membrane. We have now rephrased our statement to indicate that the dissipated membrane potential as a result of lysine deprotonation would prevent the peptide from crossing the membrane. We have removed the notion that lysine-containing peptides get stuck in membranes as this would unlikely happen based on this reviewer’s explanations. We have also added the MacCallum study to the references indicating that lysines get deprotonated in membranes.

Page 22: "…water pores created by megapolarization have a diameter of about 2(-5) nm. Molecules larger than 2 nm are therefore less efficiently transported through these water pores. Polyarginine peptides of 20 amino acids or more have a predicted spherical diameter greater than 2 nm and would travel less efficiency through the water pores than shorter peptides. The efficiency of direct translocation is therefore likely modulated by the number of positive charges of the peptide and the size of the peptides (not mentioning the role of the secondary structures adopted by the CPPs)."I agree with the general statement that the uptake of large molecules could be hindered or even prevented by the limited size of the pore size. However, the PEPFOLD-3 server, which the authors used for the prediction of peptide structures fixed in all simulations, predicts helical structure for polyarginine peptides with lengths 9 and even 20. Such helix has a diameter smaller than 2 nm along the helical axes and thus can easily get through the pore. Therefore, the presented data/discussion is not consistent with simulations.

We agree with the reviewer that larger molecules could use the water pores if they adopt thread-like structures but the general concept remains that since the pore has a given diameter, molecules with increasing sizes will be less and less likely on average to translocate via the water pore. To speculate less on the size that polyarginine peptides adopt when they translocate into cells we have now removed the mention of specific diameters. We have just kept the notion mentioned above, namely that size will influence the translocation of molecules across the water pores and that this might also apply to polyarginine peptides at some point.

Finally, the authors correctly state that arginine residues could be replaced by tryptophan residues in CPPs leading to similar translocation in cells. This possibility clearly demonstrates that there is more into the CPP translocation mechanism than the electrostatic membrane potential. The used explanation "It appears that loss of positive charges that contribute to water pore formation can be compensated by acquisition of strengthened lipid interactions when arginine residues are replaced with tryptophan residues." is strange because the use of more residues with less hydrophobic character would then have the same effect (e.g., two phenylalanine residues instead of arginine or tryptophan). In addition, the membrane potential does not explain the experiments on GUVs. Therefore, despite a nice demonstration of the importance of the membrane potential for the CPP translocation into cells, the mechanism seems to be more complex.

We understand this reviewer concern. To be less speculative we have removed the quoted sentence and also added a last sentence in this paragraph indicating indeed that the direct translocation mechanisms is unlikely explained solely by the electric characteristics of the CPPs.

References

1. Wallbrecher, R., Ackels, T., Olea, R. A., Klein, M. J., Caillon, L., Schiller, J., Bovee-Geurts, P. H., van Kuppevelt, T. H., Ulrich, A. S., Spehr, M., Adjobo-Hermans, M. J. W. & Brock, R. Membrane permeation of arginine-rich cell-penetrating peptides independent of transmembrane potential as a function of lipid composition and membrane fluidity. *J Control Release* 256, 68-78, doi:10.1016/j.jconrel.2017.04.013 (2017).

2. Zhang, X., Jin, Y., Plummer, M. R., Pooyan, S., Gunaseelan, S. & Sinko, P. J. Endocytosis and membrane potential are required for HeLa cell uptake of R.I.-CKTat9, a retro-inverso Tat cell penetrating peptide. *Mol Pharm* 6, 836-848, doi:10.1021/mp800121f (2009).

3. Herce, H. D., Garcia, A. E. & Cardoso, M. C. Fundamental molecular mechanism for the cellular uptake of guanidinium-rich molecules. *J Am Chem Soc* 136, 17459-17467, doi:10.1021/ja507790z (2014).

4. Ziegler, A., Nervi, P., Dürrenberger, M. & Seelig, J. The Cationic Cell-Penetrating Peptide CPPTAT Derived from the HIV-1 Protein TAT Is Rapidly Transported into Living Fibroblasts:  Optical, Biophysical, and Metabolic Evidence. *Biochemistry* 44, 138-148, doi:10.1021/bi0491604 (2005).

5. Duchardt, F., Fotin-Mleczek, M., Schwarz, H., Fischer, R. & Brock, R. A comprehensive model for the cellular uptake of cationic cell-penetrating peptides. *Traffic* 8, 848-866, doi:10.1111/j.1600-0854.2007.00572.x (2007).

6. Potocky, T. B., Menon, A. K. & Gellman, S. H. Cytoplasmic and nuclear delivery of a TAT-derived peptide and a beta-peptide after endocytic uptake into HeLa cells. *J Biol Chem* 278, 50188-50194, doi:10.1074/jbc.M308719200 (2003).

7. Ter-Avetisyan, G., Tunnemann, G., Nowak, D., Nitschke, M., Herrmann, A., Drab, M. & Cardoso, M. C. Cell entry of arginine-rich peptides is independent of endocytosis. *J Biol Chem* 284, 3370-3378, doi:10.1074/jbc.M805550200 (2009).

8. Lazzari-Dean, J. R., Gest, A. M. & Miller, E. W. Optical estimation of absolute membrane potential using fluorescence lifetime imaging. *Elife* 8, doi:10.7554/eLife.44522 (2019).

9. Marrink, S. J. & Tieleman, D. P. Perspective on the Martini model. *Chem Soc Rev* 42, 6801-6822, doi:10.1039/c3cs60093a (2013).

10. Marrink, S. J., Corradi, V., Souza, P. C. T., Ingolfsson, H. I., Tieleman, D. P. & Sansom, M. S. P. Computational Modeling of Realistic Cell Membranes. *Chem Rev* 119, 6184-6226, doi:10.1021/acs.chemrev.8b00460 (2019).

11. Monticelli, L., Kandasamy, S. K., Periole, X., Larson, R. G., Tieleman, D. P. & Marrink, S. J. The MARTINI Coarse-Grained Force Field: Extension to Proteins. *J Chem Theory Comput* 4, 819-834, doi:10.1021/ct700324x (2008).

12. Lamiable, A., Thevenet, P., Rey, J., Vavrusa, M., Derreumaux, P. & Tuffery, P. PEP-FOLD3: faster de novo structure prediction for linear peptides in solution and in complex. *Nucleic Acids Res* 44, W449-454, doi:10.1093/nar/gkw329 (2016).

13. Grasso, G., Muscat, S., Rebella, M., Morbiducci, U., Audenino, A., Danani, A. & Deriu, M. A. Cell penetrating peptide modulation of membrane biomechanics by Molecular dynamics. *J Biomech* 73, 137-144, doi:10.1016/j.jbiomech.2018.03.036 (2018).

14. Grasso, G., Deriu, M. A., Prat, M., Rimondini, L., Verne, E., Follenzi, A. & Danani, A. Cell Penetrating Peptide Adsorption on Magnetite and Silica Surfaces: A Computational Investigation. *J Phys Chem B* 119, 8239-8246, doi:10.1021/jp512782e (2015).

15. Koch, S., Exterkate, M., Lopez, C. A., Patro, M., Marrink, S. J. & Driessen, A. J. M. Two distinct anionic phospholipid-dependent events involved in SecA-mediated protein translocation. *Biochim Biophys Acta Biomembr* 1861, 183035, doi:10.1016/j.bbamem.2019.183035 (2019).

16. Sengupta, D. & Marrink, S. J. Lipid-mediated interactions tune the association of glycophorin A helix and its disruptive mutants in membranes. *Phys Chem Chem Phys* 12, 12987-12996, doi:10.1039/c0cp00101e (2010).

17. Periole, X., Cavalli, M., Marrink, S. J. & Ceruso, M. A. Combining an Elastic Network With a Coarse-Grained Molecular Force Field: Structure, Dynamics, and Intermolecular Recognition. *J Chem Theory Comput* 5, 2531-2543, doi:10.1021/ct9002114 (2009).

18. Castillo, N., Monticelli, L., Barnoud, J. & Tieleman, D. P. Free energy of WALP23 dimer association in DMPC, DPPC, and DOPC bilayers. *Chem Phys Lipids* 169, 95-105, doi:10.1016/j.chemphyslip.2013.02.001 (2013).

19. Marrink, S. J., de Vries, A. H. & Mark, A. E. Coarse Grained Model for Semiquantitative Lipid Simulations. *The Journal of Physical Chemistry B* 108, 750-760, doi:10.1021/jp036508g (2004).

20. Ramadurai, S., Holt, A., Schafer, L. V., Krasnikov, V. V., Rijkers, D. T., Marrink, S. J., Killian, J. A. & Poolman, B. Influence of hydrophobic mismatch and amino acid composition on the lateral diffusion of transmembrane peptides. *Biophys J* 99, 1447-1454, doi:10.1016/j.bpj.2010.05.042 (2010).

21. Periole, X., Huber, T., Marrink, S. J. & Sakmar, T. P. G protein-coupled receptors self-assemble in dynamics simulations of model bilayers. *J Am Chem Soc* 129, 10126-10132, doi:10.1021/ja0706246 (2007).

22. Monticelli, L., Tieleman, D. P. & Fuchs, P. F. Interpretation of 2H-NMR experiments on the orientation of the transmembrane helix WALP23 by computer simulations. *Biophys J* 99, 1455-1464, doi:10.1016/j.bpj.2010.05.039 (2010).

23. Marrink, S. J., Risselada, H. J., Yefimov, S., Tieleman, D. P. & de Vries, A. H. The MARTINI force field: coarse grained model for biomolecular simulations. *J Phys Chem B* 111, 7812-7824, doi:10.1021/jp071097f (2007).

24. Risselada, H. J. & Marrink, S. J. The molecular face of lipid rafts in model membranes. *Proc Natl Acad Sci U S A* 105, 17367-17372, doi:10.1073/pnas.0807527105 (2008).

25. Baron, R., Trzesniak, D., de Vries, A. H., Elsener, A., Marrink, S. J. & van Gunsteren, W. F. Comparison of thermodynamic properties of coarse-grained and atomic-level simulation models. *Chemphyschem* 8, 452-461, doi:10.1002/cphc.200600658 (2007).

26. Marrink, S. J. & Mark, A. E. Molecular view of hexagonal phase formation in phospholipid membranes. *Biophys J* 87, 3894-3900, doi:10.1529/biophysj.104.048710 (2004).

27. Marrink, S. J., Risselada, J. & Mark, A. E. Simulation of gel phase formation and melting in lipid bilayers using a coarse grained model. *Chem Phys Lipids* 135, 223-244, doi:10.1016/j.chemphyslip.2005.03.001 (2005).

28. Faller, R. & Marrink, S. J. Simulation of domain formation in DLPC-DSPC mixed bilayers. *Langmuir* 20, 7686-7693, doi:10.1021/la0492759 (2004).

29. Yesylevskyy, S. O., Schafer, L. V., Sengupta, D. & Marrink, S. J. Polarizable water model for the coarse-grained MARTINI force field. *PLoS Comput Biol* 6, e1000810, doi:10.1371/journal.pcbi.1000810 (2010).

30. Kirsch, S. A. & Bockmann, R. A. Membrane pore formation in atomistic and coarse-grained simulations. *Biochim Biophys Acta* 1858, 2266-2277, doi:10.1016/j.bbamem.2015.12.031 (2016).

31. Serulla, M., Ichim, G., Stojceski, F., Grasso, G., Afonin, S., Heulot, M., Schober, T., Roth, R., Godefroy, C., Milhiet, P. E., Das, K., Garcia-Saez, A. J., Danani, A. & Widmann, C. TAT-RasGAP317-326 kills cells by targeting inner-leaflet-enriched phospholipids. *Proc Natl Acad Sci U S A*, doi:10.1073/pnas.2014108117 (2020).

32. Armstrong, C. T., Mason, P. E., Anderson, J. L. & Dempsey, C. E. Arginine side chain interactions and the role of arginine as a gating charge carrier in voltage sensitive ion channels. *Sci Rep* 6, 21759, doi:10.1038/srep21759 (2016).

33. Li, L., Vorobyov, I. & Allen, T. W. The different interactions of lysine and arginine side chains with lipid membranes. *J Phys Chem B* 117, 11906-11920, doi:10.1021/jp405418y (2013).

34. Mitchell, D. J., Kim, D. T., Steinman, L., Fathman, C. G. & Rothbard, J. B. Polyarginine enters cells more efficiently than other polycationic homopolymers. *J Pept Res* 56, 318-325, doi:10.1034/j.1399-3011.2000.00723.x (2000).

35. Kosuge, M., Takeuchi, T., Nakase, I., Jones, A. T. & Futaki, S. Cellular Internalization and Distribution of Arginine-Rich Peptides as a Function of Extracellular Peptide Concentration, Serum, and Plasma Membrane Associated Proteoglycans. *Bioconjugate Chemistry* 19, 656-664, doi:10.1021/bc700289w (2008).